# Evaluation of the peer leadership for physical literacy intervention: A cluster randomized controlled trial

Ryan M. Hulteen[1]\*, David R. Lubans[2,3,4], Ryan E. Rhodes[5], Guy Faulkner[6], Yan Liu[7], Patti-Jean Naylor[5], Nicole Nathan[3,8,9,10], Katrina J. Waldhauser[6], Colin M. Wierts[6], Mark R. Beauchamp[6]

1 School of Kinesiology, Louisiana State University, Baton Rouge, Louisiana, United States of America, 2 School of Education, Centre for Active Living and Learning, University of Newcastle, Callaghan, New South Wales, Australia, 3 Hunter Medical Research Institute, New Lambton Heights, New South Wales, Australia, 4 Faculty of Sport and Health Sciences, University of Jyväskylä, Jyväskylä, Finland, 5 School of Exercise Science, Physical and Health Education, University of Victoria, Victoria, British Columbia, Canada, 6 School of Kinesiology, The University of British Columbia, Vancouver, British Columbia, Canada, 7 Department of Psychology, Carleton University, Ottawa, Ontario, Canada, 8 Hunter New England Population Health, Hunter New England Area Health Service, Newcastle, Wallsend, New South Wales, Australia, 9 School of Medicine and Public Health, The University of Newcastle, University Drive, Callaghan, New South Wales, Australia, 10 Priority Research Centre for Health Behaviour, The University of Newcastle, University Drive, Callaghan, New South Wales, Australia

\* rhulteen@lsu.edu

**Data Availability Statement:** All relevant data are within the manuscript and its Supporting Information files.

## Abstract

### Purpose

The purpose of this research was to develop, implement, and test the efficacy of a theory-driven, evidence-informed peer leadership program for elementary school students (Grade 6 and 7; age 11–12 years) and the Grade 3/4 students with whom they were partnered. The primary outcome was teacher ratings of their Grade 6/7 students' transformational leadership behaviors. Secondary outcomes included: Grade 6/7 students' leadership self-efficacy, as well as Grade 3/4 motivation, perceived competence, general self-concept, fundamental movement skills, school-day physical activity, and program adherence, and program evaluation.

### Methods

We conducted a two-arm cluster randomized controlled trial. In 2019, 6 schools comprising 7 teachers, 132 leaders, and 227 grade 3 and 4 students were randomly allocated to the intervention or waitlist control conditions. Intervention teachers took part in a half-day workshop (January 2019), delivered 7 x 40 minute lessons to Grade 6/7 peer leaders (February and March 2019), and these peer leaders subsequently ran a ten-week physical literacy development program for Grade 3/4 students (2x30 minutes sessions per week). Waitlist-control students followed their usual routines. Assessments were conducted at baseline (January 2019) and immediately post-intervention (June 2019).

**Funding:** Author M.B. received funding to conduct this project through the Social Sciences and Humanities Research Council of Canada (SSHRC; https://www.sshrc-crsh.gc.ca/home-accueil-eng. aspx) Insights Grant #435-2017-0268. Author DRL is supported by a National Health and Medical Research Council Senior Research Fellowship (APP1154507; https://www.nhmrc.gov.au/). The funding bodies, in both instances, had no role in the design, data collection, analysis, interpretation of data, or the writing of the manuscript.

**Competing interests:** The authors have declared that no competing interests exist.

## Results

The intervention had no significant effect on teacher ratings of their students' transformational leadership ($b = 0.201$, p = .272) after controlling for baseline and gender. There was no significant condition effect for Grade 6/7 student rated transformation leadership ($b = 0.077$, p = .569) or leadership self-efficacy ($b = 3.747$, p = .186) while controlling for baseline and gender. There were null findings for all outcomes related to Grade 3 and 4 students.

## Discussion

Adaptions to the delivery mechanism were not effective in increasing leadership skills of older students or components of physical literacy in younger Grade 3/4 students. However, teacher self-reported adherence to the intervention delivery was high.

## Trial registration

This trial was registered on December 19[th], 2018 with Clinicaltrials.gov (NCT03783767), https://clinicaltrials.gov/ct2/show/NCT03783767.

## Introduction

Schools have been identified as ideal settings to promote various life skills and health behaviors in children and youth [1]. Within elementary schools, there are both opportunities and curriculum requirements to schedule time during the school week to teach relevant life skills and health behaviors [2]. One important life skill that represents a viable target for intervention is *leadership*. Broadly conceived, leadership is concerned with the behavioral processes through which one person influences a group of others toward attaining a specific set of goals or objectives [3]. Effective displays of leadership, by children, have the potential to affect the learning climate within the class context, other facets of school life (e.g., sports, extra-curricular activities), and potentially beyond (e.g., early adulthood). In spite of the importance of developing leadership among children, there is a distinct paucity of theory-driven and/or evidence informed physical activity leadership interventions/programs designed to develop leadership [4–6].

Although many leadership frameworks exist, transformational leadership theory [7] has been the most widely used across various settings, including schools. Transformational leadership is conceptualized as involving the demonstration of behaviors that empower and inspire others, transcend one's own self-interests, and give others the confidence to achieve higher levels of functioning [8]. Transformational leadership involves four conceptually distinct, but related, dimensions: (a) idealized influence (fosters trust and respect, acts as a role model), (b) inspirational motivation (displays optimism, enthusiasm, and communicates high expectations of others), (c) intellectual stimulation (encourages others to see issues from multiple perspectives), and (d) individualized consideration (recognizes and acts on the psychological and physical needs of others) [7, 8]. In a range of life contexts (e.g., sport, business, education), observational studies shown positive associations between displays of transformational leadership and follower motivation [9], self-efficacy [10], wellbeing [11], and engagement [12, 13].

A recent review of peer-delivered interventions that sought to promote physical activity behaviors revealed a distinct absence of studies that sought to assess the behaviors of peer-leaders (i.e., leadership behaviors) that were targeted within the respective interventions [6].

Indeed, the only study in this review that assessed both leadership behaviors among peer leaders, as well as the physical activity behaviors of those being led was the Australian GLASS trial (i.e., pilot study for the current randomized controlled trial) [14]. Specifically, the Great Leaders Active StudentS (GLASS) non-randomized trial sought to develop the transformational leadership skills of older students (Grade 6 *peer-leaders*; 11–12 years of age) and increase physical activity levels and fundamental movement skill (FMS) competency among younger students in Kindergarten, Grade 1 and Grade 2 [14]. This 10-week, peer-leadership intervention was directly informed by tenets of transformational leadership theory [7] and previous movement skill interventions [15–17]. It was developed to enhance the leadership behaviors of elementary school children, who were subsequently tasked with developing younger children's FMS competency [14]. FMS represent the building blocks required for children and youth to engage in lifelong health-enhancing physical activities [18, 19], and represent a key behavioral component of *physical literacy* [20].

Physical literacy is defined as "motivation, confidence, physical competence, knowledge, and understanding to value and take responsibility for engagement in physical activities for life" [20]. Physical literacy thus takes a more holistic perspective in understanding the attributes that contribute to one's capability to be physically active across the lifespan. Multiple interventions have focused on individual components of physical literacy from a physical, psychological, or social perspective [21]. However, targeting multiple components of physical literacy within a single intervention may be more advantageous, as it will help elucidate those factor(s) that best assist an individual in being physically active. For example, a recent systematic review and meta-analysis of physical literacy interventions demonstrated that there were significant treatment effects for interventions when all physical literacy outcomes were combined [21]. Yet, these results are more nuanced with interventions exerting differential effects on the varying outcome categories. This same review describes that there appears to be the strongest evidence for improving physical competence, while psychological outcomes (e.g., motivation, confidence) are harder to improve through interventions [21].

Findings from this systematic review [21] align with findings from our team's previous GLASS pilot trial. Indeed, our previous pilot intervention resulted in large effects for teacher rated leadership effectiveness among Grade 6 students ($d = 1.09$) as well as the object control movement skill competency of younger students ($d = 0.95$) [14]. The study did not result in a statistically significant difference in terms of school day physical activity ($d = 0.29$; $p = 0.313$). However, this pilot study was not without its limitations. Specifically, the GLASS trial used a non-randomized study design, was only implemented in one school, used pedometers as a measure of physical activity, and was implemented with the youngest children in primary school (i.e., Kindergarden, Grade 1 and 2). Further, the intervention delivery was very labor intensive. Leadership development training was delivered directly by a researcher to the peer-leaders. Additional resources including providing peer leaders with equipment packs to use during their lessons, FMS training, a one-hour booster training midway through the intervention, as well as handbooks and lesson plans, and three weeks where observations and feedback were provided on their teaching were also provided by the researchers in this study. Ultimately, larger scale studies are needed that involve a more scalable delivery model. To build upon past success in the GLASS trial [14], the current trial tested the efficacy of a *Peer Leadership for Physical Literacy* (PLPL) intervention. The key goal of the PLPL intervention was to develop both leadership behaviors (primary outcome) and components of physical literacy (secondary outcomes) among students in Canadian elementary schools. We used a cluster randomized controlled trial design with an expanded number of schools and a different delivery mechanism. In the current trial, adaptations were made to enhance the potential scalability of the intervention [22, 23] by ensuring intervention components or implementation strategies

were less resource intensive. Thus, teachers were provided with the necessary training and curriculum resources to deliver the leadership training/education intervention directly to peer leaders. If efficacious, this approach would lend itself to broader intervention delivery and reach [24, 25].

## Aims and hypotheses

The aims of this study were to develop, implement, and test the efficacy of a theory-driven, scalable, evidence-based peer leadership program for elementary school students (Grade 6/7; age 11–12 years) in relation to (a) their own leadership skills (primary outcome) and their leadership self-efficacy (i.e., confidence to lead; secondary outcome), as well as (b) components of physical literacy of younger (Grade 3/4; age 8–9 years) students with whom they were partnered for 10 weeks (secondary outcomes). We hypothesized that the intervention would result in greater use of peer leadership behaviors by Grade 6/7 students as rated by their teacher, compared to control peers (primary research question). As secondary research questions, we also hypothesized that the intervention would result in higher levels of peer leaders' confidence to lead, and as well as increases in outcomes related to components of physical literacy in Grade 3/4 students' when compared to control students.

## Methods

### Design

The PLPL intervention was evaluated using a two-arm cluster randomized controlled trial.

The study was approved by the Behavioural Research Ethics Board at The University of British Columbia (H18-00141), with School Board approval also obtained from four School Boards/Districts in the Lower Mainland of British Columbia. Prior to enrollment in the study, informed consent was obtained from parents/guardians and teachers, as well as signed assent from all children. The trial was registered with ClinicalTrials.gov (#NCT03783767; Registered December 21, 2018). The design, conduct, and reporting of the study adhered to the Consolidated Standards of Reporting Trials (CONSORT) extension for cluster randomized controlled trials [26] (See Fig 1), Standard Protocol Items: Recommendations for Interventional Trials (SPIRIT) guidelines [27] (See S1 Fig for SPIRIT diagram), and the Template for Intervention Description and Replication (TIDieR) checklist (See S1 Table) [28].

To effectively manage the study, the trial was designed to run in two cohorts, with the first cohort taking place between January and June 2019, and the second cohort running between January and June 2020 (see Fig 1). In each cohort, baseline measures were collected in January, after which schools were randomized to either intervention or wait-list control conditions. Teachers of Grade 6/7 students in the intervention condition received training and curriculum resources (February) to deliver a leadership development module to Grade 6/7 students (hereafter referred to as peer leaders). From April to June peer leaders delivered a 10-week movement skills program to their younger Grade 3/4 counterparts. Post-test measures occurred at the end of the 10-week program (June) for both intervention and wait-list control participants who engaged in the 'usual practice' curriculum.

### Changes to trial design

In March of 2020 study recruitment was terminated early. While the PLPL trial was originally designed to run in two cohorts, shutdowns due to COVID-19 necessitated an early termination of recruitment. Prior to this shutdown, we were able to fully complete Cohort 1 of this study in 2019. Thus, this manuscript reflects only data from Cohort 1. We did not collect data

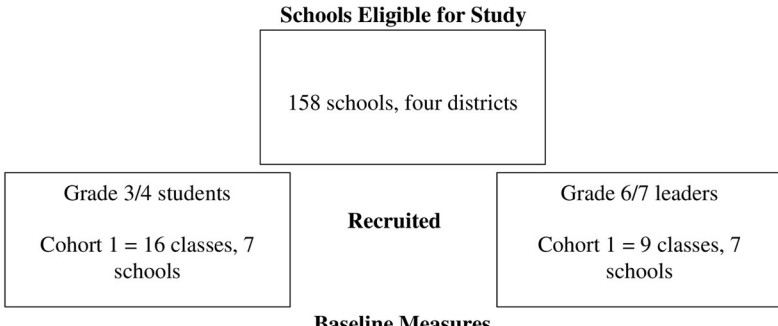

**Fig 1. CONSORT flow diagram here**[***].

from the second cohort once restrictions were eased due to the length of time the study had been paused and the substantive changes in physical activity due to COVID-19 [29, 30] that potentially rendered a second cohort incomparable.

## Participants

Elementary schools within four school districts in the Lower Mainland of British Columbia, Canada were invited to participate in this project. Recruitment for Cohort 1 took place between September and December 2018. All principals at eligible schools (public schools within the four districts for which district approval was obtained) were contacted by email for initial interest in the study. Following an expression of interest from a principal, a meeting with the principal and eligible teachers was conducted. After the meeting, schools determined whether they wanted to participate, and which classes would be involved. Schools were asked

to provide at least one class of older students (Grade 6/7; ages 12–13), as well as up to three classes of younger students (ages 8–10). Grade 7 represents the last year of elementary school in the target school districts. Split grade classes are common thus the class of older students could have been all Grade 7 students or both Grade 6 and 7 students. For younger students, classes were also split, but only Grade 3 and 4 students took part in the study (enrolled in all Grade 4, 3/4 or 4/5 split classes). The only exclusion criterion at the student level was a basic working knowledge of English. Active parental consent, as well as child assent, were obtained before enrollment in the study.

## Blinding and randomization

After baseline assessments (January 2019) schools were randomized (by a researcher not involved in the current study using a computer-based, random number generator) with an allocation ratio of 1:1 to either intervention or waitlist-control conditions. School allocation to the intervention or control group (i.e., usual care) was blinded to all data collectors (aside from the project manager responsible for correct delivery of intervention materials). Schools allocated to the wait-list control group received the PLPL program in full during the following academic year (Fall 2019), but did not contribute data to the trial when receiving the PLPL program. Notification of group allocation to either group was completed by the project manager via email.

## Intervention

The PLPL trial was implemented in two phases. The first phase involved the development of leadership among Grade 6/7 peer leaders; referred to as the 'Peer-Leadership Development Program' section of the intervention. The second phase involved the Grade 6/7 peer leaders delivering a 10-week movement skills program to the younger Grade 3/4 students; referred to as the 'Physical Literacy Development Program' of the overall intervention. Both sections drew directly from our previous work [14], as well as previous intervention work within school settings that involved a train-the-teacher delivery model [15, 31]. Two key changes were made to the current trial compared to our pilot study. First, the target age group of younger students was different, shifting from Kindergarten, Grade 1 and 2 in our previous non-randomized trial to Grade 3 and 4 students in this efficacy trial. Secondly, previously where a research team delivered all content to peer leaders, a train-the-trainer approach was adopted and the research team delivered all content to Grade 6/7 teachers through a half-day workshop at the last author's university. These teachers then delivered the content to their students. This change in delivery mechanism was designed with intervention scalability and sustainability foregrounded. Specifically, having teachers deliver programs within their own classes are more likely to be sustained and delivered at scale compared to an external research team that would be required to deliver a program across multiple schools on an ongoing basis.

As an incentive for participation, schools (in both intervention and wait-list control conditions) were provided with an assortment of developmentally appropriate sports equipment (e.g., basketballs, soccer balls, baseball batting tees, cones, beanbags) to be used during the movement skills program. The value of these equipment packs was approximately $250 each. Schools retained this equipment after the study. For participation at the individual student level, all peer leaders who completed questionnaires and Grade 3/4 students who completed questionnaires and movement skill measures (described below) at baseline and post-test received $10. Grade 3/4 students who also completed accelerometry measures at both time points received an additional $20.

## Peer-leadership development program (intervention part I)

The Peer-Leadership Development Program (see S2 Table) is guided by the conceptual framework initially developed by Kelloway and Barling [32] and extended by Beauchamp and Morton [33]. This framework involves (a) presentation of transformational leadership behavioral principles, (b) demonstration of what those behaviors look like in practice, using real-world examples, (c) providing opportunities to practice those behaviors, (d) receiving feedback on the implementation of those strategies, and (e) development of self-regulatory strategies to support sustained implementation [12]. The program was delivered in age-appropriate terms [14]. We refer to the four dimensions of transformational leadership as 'role modelling' (idealized influence), 'motivating others' (inspirational motivation), 'considering others' (individualized consideration), and 'helping students to think' (intellectual stimulation). At the end of the peer-leadership development module, peer leaders introduced themselves to their allotted Grade 3/4 students.

One month after baseline measures were collected, Grade 6/7 teachers from the intervention schools attended a half-day workshop (4 hours) in which teachers were provided with the curriculum materials, resources, and lesson plans to be subsequently delivered to their students (see S2 Table). In the workshop, an intervention facilitator (MB; who has 10 years of school-based intervention experience) introduced and explained all curriculum materials to teachers. Teachers were encouraged to discuss and work through any challenges with curriculum delivery. Teachers were provided with a manual to facilitate (a) the delivery of the peer-leadership development program, and (b) peer leaders' subsequent delivery of the 10-week movement skills program. At the end of the workshop, teachers worked with the intervention facilitator to schedule the delivery of the in-school, peer-leadership program.

Following the workshop, teachers delivered four 40-minute lessons on leadership over a 6-week period (classes 1–4; see S2 Table), followed by an experiential component (classes 5–7; see S2 Table). The four 40-minute lessons (i.e., classes 1–4) were designed to explain (a) what exceptional leadership looks like (i.e., the behaviors and attributes of great leaders), (b) the benefits of movement skills, (c) the key teaching points used to explain each key movement skill incorporated in this program, and (d) how peer leaders will use their leadership skills to deliver a 10-week movement skills program to Grade 3/4 students. In the experiential component (i.e., classes 5–7), Grade 6/7 peer leaders practiced delivering aspects of the movement skills program (3 X 40 min classes) to each other and received feedback from each other and their teacher, before they delivered the 10-week movement skills program to the Grade 3/4 students (see Intervention Part II below). If teachers felt that students needed additional classes to practice their teaching before implementing the 10-week program, they were able to add practice classes as needed.

## Physical literacy development program (intervention part II)

The Grade 6/7 peer leaders delivered two 30-minute movement skill sessions per week for 10-weeks, at a convenient time for the schools (e.g., beginning of school, lunch, during younger students' physical education classes). To provide further flexibility, lessons could occur in a gymnasium or on an outdoor sports field. The 10-week program provided a maximum of 600 minutes of program delivery to younger students. Within intervention schools, a class of peer leaders (~25–30 students) was divided into three groups. Thus, between 6–9 peer leaders were allocated to deliver movement skills sessions with one entire younger class (~20–25 students) for the duration of the 10-week movement skills program. Peer leaders remained with the same class throughout the 10-week period. Peer leaders worked in pairs to teach one section of each class (i.e., warm-up, demonstrate the movement skill, skill practice, and set-up and

oversee the game). Thus, 2–3 leaders would be responsible for teaching the warm-up, then a different 2–3 leaders would teach skill practice, and then a different 2–3 leaders would teach the game. While lesson components were being completed all leaders were able to provide feedback to younger students as necessary. At the beginning of each week, peer leaders planned their respective lessons for which they were responsible. In addition to displaying leadership by teaching specific sections of the movement skills sessions, all peer leaders were encouraged to display effective leadership (e.g., act as effective role models) by taking part in the activities and games along with the Grade 3 and 4 students they taught.

Each movement skill session focused on one of six object control skills (i.e., catching, over-hand throwing, underhand throwing, kicking, dribbling, and a two-handed strike (with a base-ball bat)). Each of these skills was taught between three (catch, overarm throw, two-handed strike, dribble) and four times (underarm throw, kick) throughout the duration of the 10-week program. While specific key teaching points were provided for all skills, as well as games that incorporated these skills, the leaders had the freedom to plan their lessons based on their own experiences. That is, a general lesson/structure was provided by the research team, but leaders could design the specific session plan.

To further support the intervention schools in administering the movement skill program to Grade 3/4 students, during the first three weeks of the program, peer leaders were observed by trained research assistants and provided with feedback with regard to (a) the core components of each class (i.e., warm-up, skill development, skill application, and closure), and (b) their leadership behaviors over the course of the class based on the four dimensions of transformational leadership described above.

## Primary outcome

*Teacher-rated transformational leadership*: The leadership behaviors of Grade 6/7 peer leaders were assessed by their teacher using an adapted version of the Transformational Teaching Questionnaire [33]. Teachers were asked to rate the behaviors of each peer leader within their respective Grade 6/7 class with a four-item measure that assessed students' displays of individualized consideration, inspirational motivation, intellectual stimulation, and idealized influence. Responses were provided on a five-point scale anchored from 0 ("Not at all") to 4 ("Frequently"), with "The peer leader that I'm rating . . ." provided as the stem. Exemplar items include "Behaves as someone that other students can trust" (idealized influence) and "Is enthusiastic about what other students are capable of achieving" (inspirational motivation). Scores derived from this abbreviated measure of the Transformational Teaching Questionnaire displayed good internal consistency ($\alpha \geq 0.92$–0.93; See S3 Table) in the current sample. The full questionnaire can be viewed in the S1 File.

## Secondary outcomes

**Grade 6/7 students.** *Student-rated transformational leadership*: In addition to Grade 6/7 teachers rating their students' leadership behaviors, students also self-rated the same behaviors, using the same abbreviated measure of the Transformational Teaching Questionnaire as used by the teachers. In this instance, the stem directs students to rate their own behaviors (e.g., "I behave as someone that other students can trust"). The same 5-point scale and anchors, as per the teacher-rating version, were used for the student self-rating version. Results obtained from this measure displayed adequate internal consistency ($\alpha \geq 0.65$–0.70). See S3 Table.

*Leadership self-efficacy*: Peer leaders' leadership self-efficacy was assessed using an 11-item measure based on standard protocols for assessing self-efficacy beliefs [34], by asking them "how confident are you that you can. . ." perform the key leadership behaviors being targeted

in the intervention. Responses to items were provided on a 0–100 scale (at 10-point increments) anchored by 0% ("no confidence"), 50% ("somewhat confident"), and 100% ("completely confident"). To ensure conceptual congruence between the target (leadership) behavior [35] and the efficacy beliefs being assessed, items from the Transformational Teaching Questionnaire were adapted to assess peer-leaders' confidence to display the relevant transformational leadership behaviors (e.g., ". . .behave as someone that other students can trust"). Items were also adapted from a previous questionnaire [36] with exemplar items including ". . .be a role model to other students", ". . .teach physical activity skills to other students", ". . .help students feel safe about joining in". As per recent recommendations [37], the prefix "If you really wanted to" was placed before "how confident are you that you can. . ." in order to hold motivation constant and derive more precise measures of self-efficacy. Measures derived from this instrument were found to display acceptable internal consistency ($\alpha \geq 0.93$) in our sample. See S3 Table.

**Grade 3/4 students.** *Self-determined motivation*: Motivation for being physically active among Grade 3/4 students was assessed using the intrinsic motivation measure developed by Sebire and colleagues [38] for specific use with children aged 7–11 years in school physical education settings. Each item begins with the stem "I am active because. . .". Example items include: "being active is fun", "I like being active", and "it is important to me to do active things". This six-item questionnaire was rated by students on a scale anchored from one (not true for me) to five (very true for me). Scores derived from this instrument have been found to display sound reliability ($\alpha \geq 0.86$–$0.89$; See S4 Table) in the current sample, as well as factorial validity in relation to the other motivation regulations subsumed within Self-Determination Theory [39].

*Perceived competence*: Perceived competence was assessed through a five-item subscale previously developed by Sebire and colleagues [38]. The original scale asked questions about perceived competence specifically in physical education. However, these questions were adapted to reflect a broader focus of perceived competence as it relates to active games, playing outside, and participating in sports. Example items include: "When it comes to playing active games, I think I am pretty good", "I think I do well compared to other children", and "After working at a new activity for a while, I feel that I can do it pretty well". Each item was rated on a scale that was anchored from 1 (not like me at all) to 5 (really like me). Scores derived from this measure demonstrated sound reliability ($\alpha \geq 0.84$–$0.87$) in the current sample. See S4 Table.

*Self-concept*: Self-concept among Grade 3/4 students was assessed using select subscales (coordination, activity, sport, global physical, and global esteem; 20 items) derived from the Physical Self-Description Questionnaire-Short Version [40]. Students were asked to rate how correct a statement is about them. Exemplar items included: "I feel confident when doing coordinated movements" (coordination), "I do physically active things at least three times a week" (activity), "I have good sports skills" (sport), "Physically, I feel good about myself" (global physical), and "Overall, I have a lot to be proud of" (global esteem). Each item is rated on a scale from 1 (false) to 5 (true). Scores derived from the Physical Self-Description Questionnaire-Short Version were found to display sound internal consistency ($\alpha \geq 0.93$–$0.95$) in our sample. See S4 Table. Previous work has supported the factorial validity (e.g., comparative fit index = 0.97) and predictive utility of results obtained from this measure [40].

*FMS*: Three FMS (throw, kick, catch) were assessed as a measure of object control skill competence. Process scores for kicking and throwing were assessed using criteria from the Test of Gross Motor Development-3rd edition [41]. Process scores for catching were not recorded, as catching was completed in a dual-task format making it inappropriate to use criteria from the Test of Gross Motor Development-3rd edition. Scores derived from these measures have shown adequate factorial validity (i.e., one factor model for FMS competence; ($\chi2$ (65) =

327.61, p < .001, comparative fit index = .95, Tucker-Lewis index = .94, root mean square error of approximation = .10) and test-retest reliability (intraclass correlation coefficient = 0.95–0.97) [41, 42]. Participants were filmed from the side and rear-view using iPads. Skill demonstrations were not provided, but practice attempts were allowed. Two research assistants, blinded to group allocation (e.g., intervention, control) completed all coding of throwing and kicking process scores. Each research assistant was provided with one hour of training by the project coordinator (who has 10+ years experience of motor skill assessment research). This training included explanation of skill criteria and practice coding. Prior to the beginning of formalized coding, both research assistants coded a unique set of 10% of all kicking and throwing videos with no overlap between coders. Research assistants had to obtain ≥ 90% agreement at the component level for both skills with the project coordinator, which was achieved.

Product scores were also assessed for kicking (maximal speed) [43, 44], overhand throwing (maximal speed) [43, 44], and catching (number of successful catches; [45]). Throw and kick speed were completed by asking participants to throw or kick a ball as hard as they could at a wall 20 feet away. Speed was measured using a Stalker Pro II radar gun. The capability to assess skills using product and process scores provides a more holistic understanding of one's level of competence and has shown moderate to strong agreement with process scores in pediatric populations [46, 47]. In accordance with previous studies using product scores and the use of the best score [43, 44, 48], we used five complete trials of both the throw and kick respectively. The dual throw-catch task required a participant to stand behind a line (approximately three times their height) away from a wall. Upon being instructed to start the task, each participant was to take a tennis ball and throw it at a blank wall (using any throw pattern they wanted) and then try to catch the ball (on the bounce or out of the air; could not trap against body) while remaining behind the line. Participants were free to move about the area as much as they wanted. The only place they could not go was in front of their designated line. If the thrown ball did not come back across the participant's line or went too far away, the participant was to run to a basket of tennis balls (6 feet away from the participant) and obtain another tennis ball and continue the task. This sequence of events was repeated as many times as possible in 30 seconds with the number of catches recorded as the final result. Throwing while performed in this task was not assessed in any way. The catching task had two trials (30 seconds each) with the best result of the two trials (most number of catches) used for the final data analyses.

*Within-school physical activity*: A subsample of Grade 3/4 students (five students per class) were randomly selected (using a random number generator by a researcher not involved in the study) and invited to wear a waist-worn GT3X or GT3X+ accelerometer on two occasions (baseline and post-test) for one school week (five days at each time point). Accelerometers were worn for five consecutive school days, only during school hours (i.e. 9am to 3pm) [2]. Trained research assistants explained the process of wearing these devices to each student and provided a demonstration of how to wear the device on the hip (above the right knee). Data were collected and stored in 30-second epochs. Valid wear time was classified as a minimum of three school days (e.g., 9am to 3pm). At least 80% wear time during the school day was needed to be classified as a valid day [49]. Any 30-minute stretch of consecutive zeroes was classified as non-wear time. Cut points proposed by Evenson were used to classify levels of moderate-to-vigorous physical activity [50].

## Program adherence

Adherence to the movement skill intervention (i.e., did each session over the 10-week movement skill period happen) was self-reported by intervention teachers to the project manager via email. These data were only collected for intervention schools.

## Post-test program evaluation

Peer leaders' appraisals of the intervention (including Parts I and II described above) were completed at post-test, using a ten-item survey which asked about (a) their enjoyment of the program, the training they received, the length of each movement skills session and being a peer leader, (b) the perceived usefulness of the resources and feedback that the research team provided, and (c) their appraisal of whether the program helped them to be a better leader and whether the Grade 3/4 students enjoyed the program. Responses were provided on a 3-point scale, whereby peer leaders were asked to provide their level of agreement with each item, with anchors including 1 ("not really"), 2 ("a little"), and 3 ("a lot").

**Power analysis.**   Since the trial involved targeted outcomes associated with both Grade 6/7 peer leaders (e.g., leadership behaviors) as well as Grade 3/4 students (e.g., movement skill competence), two sets of power analyses were conducted to determine sample size calculations for each group. We conducted the power analyses using Optimal Design Software [51].

*Primary outcome sample size parameters***:** In relation to the study's primary outcome (teacher ratings of peer leaders' transformational leadership behaviors), we considered that each group of peer leaders would include 8–10 Grade 6/7 students (equal number of boys and girls), who would work with each class of Grade 3/4 students. Thus, one Grade 6/7 class would be able to be split into three clusters and each cluster would be responsible for teaching a whole Grade 3/4 class. The intervention effect in our pilot study for peer leadership development was d = 1.09 [14]. On the basis of alpha set at .05, Power (1 - β) set at .80, cluster size of 8, ICC = .05, effect size (δ = .80; based on established criteria for a large effect; [52]) we required 23 clusters of 8. By rounding up to 24 clusters of 8, with 3 clusters of 8 coming from one Grade 6/7 class per school, this equated to 8 schools (4 intervention and 4 control), for a total sample of 192 peer leaders.

In anticipation that a medium sized effect for leadership development might be more realistic given this efficacy trial would be implemented in more schools than our pilot study (δ = 0.60), on the basis of alpha set at .05, Power (1 - β) set at .80, cluster size of 8, ICC = .05, effect size (δ = .60) we would require 39 clusters of 8. On the basis of three clusters of 8 per school (i.e., one Grade 6/7 class per school), and to be balanced across intervention and control schools, this would necessitate (by rounding up) 42 clusters of 8, which would equate to 14 schools (7 intervention and 7 control) for a total of 336 peer leaders.

*Secondary outcomes sample size parameters***:** The sample size calculations for the secondary research questions were powered based on the effect size derived for movement skill competency derived in our pilot trial (d = .95; 14). We anticipated that each class of younger students would include approximately 24 students (equal number of boys and girls) per class (approximately 30 students per class, but after accounting for anticipated enrolment). On the basis of alpha set at .05, Power (1 - β) set at .80, cluster size of 24, ICC = .05, effect size (δ = .50; based on established criteria for a medium effect; 52) we would require 24 clusters of 24. This would also align with the number of peer leadership groups (24 clusters of 8 Grade 6/7 students) delivering the program to the younger students. Based on 24 groups of 24 Grade 3/4 students, this would equate to 576 students from 8 schools (4 intervention and 4 control).

To accommodate the more conservative effect size parameters for leadership development for peer leaders (described above), which necessitate 42 clusters of 8 peer leaders (i.e., 14 schools; 7 intervention and 7 control), this would involve 42 classes of 24 Grade 3/4 students (n = 1008).

**Analyses.**   All analyses were conducted in SPSS version 27. Data were screened for patterns of missing data and outlier values. Multiple imputation of missing data was conducted using the fully conditional specification method and regression modelling for both the leaders and younger students. Ten imputed data sets were created for both the leader and younger student data sets. Values were imputed at the item and task level (for FMS) for all study variables with

missing data. Constraints (minimum, maximum, and rounding) were specified for Likert-type items and FMS to ensure all imputed values were within the appropriate numerical range.

Descriptive statistics (mean, standard deviation) were calculated for each variable at baseline and follow up (See Tables 1 and 2). Coefficient alphas were also calculated to assess internal consistency reliability for each variable at baseline and follow up (See S3 and S4 Tables). Both older and younger students were nested within their respected classrooms. Therefore, intraclass correlations (ICCs) and design effects were calculated for baseline and follow up measures of each outcome variable (See S3 and S4 Tables) to determine whether linear mixed modelling should be used to account for a random classroom effect. The ICC represents the proportion of variance attributed to the grouping structure (i.e., classrooms) compared to the total variance. The design effect represents the magnitude of the clustering effect while considering the average cluster (i.e., classroom) size, with larger numbers representing a larger clustering effect. If the design effect is larger than 2.0, linear mixed modelling with random intercept was used to account for the nesting structure of the data. Proportion of variance reduction was calculated to examine how much more variation in the outcome was accounted for by adding the treatment condition into the model. If the design effect is smaller than 2.0 at baseline, ordinary least squares (OLS) regression was used for the data analysis. Correspondingly $R^2$ square change was reported to examine the amount of additional variance explained in an outcome when treatment condition was included in the regression model, compared to the model with only baseline assessment and gender. When design effects were > 2.0 at baseline, but < 2.0 at follow up, both linear mixed modelling and OLS regressions were used. If results did not deviate between the linear mixed model and OLS regression model, the results from the OLS regression model were reported.

In both linear mixed models and regression models, the treatment condition (0 = control, 1 = intervention) effect was modelled while controlling for baseline assessment and gender (0 = female, 1 = male). To examine whether the intervention was more effective for one gender compared to the other (e.g., was the intervention more effective for females compared to males?), condition by gender interaction effects were also included in the linear mixed models and OLS regression models alongside condition, gender, and baseline. Reported effects for the linear mixed models and OLS regression models are unstandardized beta coefficients derived from the ten analysis with ten imputed data sets.

## Results

### Sample characteristics

As mentioned above, given the second cohort could not be run due to COVID-19, the results presented below only represent results for Cohort 1. Mean and standard deviation values for outcomes are presented in Tables 1 and 2. Recruitment began by contacting the principals at

**Table 1. Teacher and Grade 6/7 student outcomes at baseline and follow-up.**

| Outcome | Intervention (n = 75) | | | | Control (n = 57) | | | |
|---|---|---|---|---|---|---|---|---|
| | n | Baseline *Mean (SD)* | n | Follow-Up *Mean (SD)* | n | Baseline *Mean (SD)* | n | Follow-Up *Mean (SD)* |
| Teacher-Rated Transformational Leadership, range (0–4) | 58 | 2.96 (0.93) | 74 | 3.25 (0.86) | 53 | 2.54 (1.08) | 57 | 2.89 (0.83) |
| Student-Rated Transformational Leadership, range (0–4) | 75 | 3.34 (0.52) | 74 | 3.17 (0.63) | 57 | 3.14 (0.49) | 55 | 3.00 (0.52) |
| Leadership Self-Efficacy, range (0–100) | 75 | 82.90 (14.04) | 74 | 82.11 (15.15) | 56 | 78.57 (14.46) | 55 | 75.72 (14.77) |

SD = Standard Deviation

[a]Descriptive statistics calculated with original data, not imputed data.

**Table 2. Grade 3/4 student outcomes at baseline and follow-up.**

| Outcome | Intervention (n = 102) | | | | Control (n = 125) | | | |
|---|---|---|---|---|---|---|---|---|
| | n | Baseline *Mean (SD)* | n | Follow-Up *Mean (SD)* | n | Baseline *Mean (SD)* | n | Follow-Up *Mean (SD)* |
| Self-Determined Motivation, range (1–5) | 98 | 4.37 (0.67) | 95 | 4.35 (0.72) | 123 | 4.32 (0.68) | 119 | 4.43 (0.68) |
| Perceived Competence, range (1–5) | 97 | 4.20 (0.70) | 95 | 4.12 (0.83) | 121 | 4.13 (0.70) | 119 | 4.20 (0.71) |
| Self-Concept, range (1–6) | 92 | 4.90 (0.86) | 95 | 4.83 (0.96) | 111 | 4.94 (0.79) | 119 | 5.01 (0.85) |
| Movement Skill Competence | | | | | | | | |
| Maximal Throw Speed, m/s | 100 | 12.59 (3.43) | 95 | 14.40 (3.35) | 122 | 14.79 (3.89) | 118 | 15.72 (3.89) |
| Maximal Kick Speed, m/s | 101 | 14.65 (2.55) | 95 | 16.02 (2.97) | 122 | 15.01 (2.82) | 118 | 15.85 (3.58) |
| Throw Components, range (0–8) | 95 | 2.49 (2.09) | 95 | 3.57 (2.38) | 122 | 3.76 (2.42) | 117 | 4.23 (2.77) |
| Kick Components, range (0–8) | 101 | 4.11 (2.60) | 95 | 3.41 (2.02) | 123 | 4.41 (2.26) | 118 | 4.32 (2.00) |
| Throw-Catch Combo, # of catches | 100 | 4.30 (3.09) | 95 | 6.55 (3.32) | 121 | 5.61 (3.71) | 119 | 7.19 (3.68) |
| School Day MVPA | 32 | 27.02 (11.40) | 31 | 29.73 (12.53) | 20 | 27.83 (10.81) | 21 | 33.10 (14.25) |

SD = Standard Deviation, m/s = meters per second, MVPA = moderate-to-vigorous physical activity

[a]Descriptive statistics reported with original data, not imputed data

158 eligible schools within four school districts in the greater Vancouver area. Of the 158 schools contacts, 68 schools did not respond to any communications, 58 schools declined participation, 25 schools expressed some level of interest in the project, and ultimately 7 schools consented to participating in the study (4.4% consent rate). Fig 1 outlines the recruitment of consenting schools and classes within these schools. Notably, one consenting school dropped out after baseline data collection due to a desire to focus on a different set of curricula. The remaining 6 schools consisted of 132 Grade 6/7 leaders (57.3% female) and 227 Grade 3/4 students (42.30% female).

## Primary outcome: Teacher rated transformational leadership

The results of the linear mixed model for students' transformational leadership are displayed in Table 3. All condition*gender interaction effects were non-significant, and therefore, only the models including non-conditional effects (baseline, gender, condition) are reported. Overall, the intervention had no significant effect on teacher ratings of their student's transformational leadership ($b = 0.201$, $p = .272$) after controlling for baseline and gender.

## Secondary outcomes

**Grade 6/7 student transformational leadership and self-efficacy.** Similar to the primary outcome, there was no significant condition effect for student rated transformational leadership ($b = 0.077$, $p = .569$) or leadership self-efficacy ($b = 3.747$, $p = .186$) while controlling for baseline and gender. There was a significant gender effect for leadership self-efficacy ($b = -4.621$, $p < .05$). Girls reported higher leadership self-efficacy than boys after controlling for baseline and condition effects (see Table 3).

**Grade 3/4 psychological and FMS outcomes.** Secondary outcomes for Grade 3/4 students are presented in Table 4 (psychological outcomes) and Table 5 (FMS). There were no condition*gender interaction effects, and therefore only the models including direct effects are reported. There were no statistically significant condition effects for any of the psychological, FMS, or physical activity outcomes in Grade 3/4 students. There were also significant gender effects for throwing speed ($b = 1.414$, $p < .001$), throwing components ($b = 1.985$, $p < .001$), and the throw-catch combination task ($b = 1.123$, $p < .01$), such that boys had higher scores than girls after controlling for baseline and condition.

**Table 3. Grade 6/7 outcome changes.**

**Teacher Rated Transformational Leadership**

| Fixed Effects | Parameter Estimates | Standard Error | p-value |
|---|---|---|---|
| Intercept | 1.497 | 0.200 | < .001 |
| Baseline | 0.628 | 0.061 | < .001 |
| Gender | 0.005 | 0.112 | .967 |
| Condition | 0.201 | 0.183 | .272 |
| *Variance Reduction* | | | |
| Level 1 | 0.000 | | |
| Level 2 | 0.202 | | |

**Student Rated Transformational Leadership**

| Fixed Effects | Parameter Estimates | Standard Error | p-value |
|---|---|---|---|
| Intercept | 1.586 | 0.326 | < .001 |
| Baseline | 0.457 | 0.096 | < .001 |
| Gender | -0.073 | 0.095 | .443 |
| Condition | 0.077 | 0.136 | .569 |
| *Variance Reduction* | | | |
| Level 1 | 0.001 | | |
| Level 2 | 0.067 | | |

**Leadership Self-Efficacy**

| Fixed Effects | Parameter Estimates | Standard Error | p-value |
|---|---|---|---|
| Intercept | 25.964 | 6.468 | < .001 |
| Baseline | 0.641 | 0.073 | < .001 |
| Gender | -4.621 | 2.026 | .023 |
| Condition | 3.747 | 2.834 | .186 |
| *Variance Reduction* | | | |
| Level 1 | 0.001 | | |
| Level 2 | 0.335 | | |

**Grade 3/4 school day physical activity.** Physical activity results are presented in Table 5. Overall, both the intervention and control group increased their physical activity from baseline (intervention = 27.02 minutes, control = 29.73 minutes) to follow-up (intervention = 27.83 minutes, control = 33.10 minutes). There were significant gender effects for school-day MVPA ($b = 8.991$, $p < .01$) demonstrating that boys were more active than girls at school. There was no significant effect for condition ($b = -3.841$, $p = .212$), which demonstrates the PLPL trial did not meaningfully improve MVPA.

**Program adherence and post-test program evaluation.** Overall, the program was well received by the intervention schools. A possible 20 sessions were to be implemented as planned by each intervention school. Of the 3 intervention schools, two implemented all sessions (self-reported 20/20 sessions delivered). The third school, due to a scheduling conflict, missed one PLPL session. Thus, these students participated in 19 of the 20 sessions (i.e., 30 minutes less intervention compared to other intervention schools). Program evaluation results are presented in S5 Table. Overall, the Grade 6/7 leaders were satisfied with the leadership program. Scores ranged from 2.39 to 2.81.

## Discussion

We developed, implemented, and tested the efficacy of a theory-driven, scalable, evidence-based peer leadership program targeting the physical literacy of elementary school students to

**Table 4. Grade 3/4 psychological outcomes.**

**Self-Determined Motivation**

| Fixed Effects | Parameter Estimates | Standard Error | p-value |
|---|---|---|---|
| Intercept | 2.276 | 0.276 | < .001 |
| Baseline | 0.476 | 0.062 | < .001 |
| Gender | 0.109 | 0.086 | .203 |
| Condition | -0.102 | 0.086 | .236 |
| $R^2$ | 0.228 | | |
| $\Delta R^2$ | 0.007 | | |

**Perceived Competence**

| Fixed Effects | Parameter Estimates | Standard Error | p-value |
|---|---|---|---|
| Intercept | 1.502 | 0.259 | < .001 |
| Baseline | 0.640 | 0.062 | < .001 |
| Gender | 0.049 | 0.085 | .563 |
| Condition | -0.116 | 0.086 | .181 |
| $R^2$ | 0.361 | | |
| $\Delta R^2$ | 0.007 | | |

**Self-Concept**

| Fixed Effects | Parameter Estimates | Standard Error | p-value |
|---|---|---|---|
| Intercept | 1.982 | 0.285 | < .001 |
| Baseline | 0.596 | 0.058 | < .001 |
| Gender | 0.123 | 0.102 | .228 |
| Condition | -0.143 | 0.100 | .153 |
| $R^2$ | 0.347 | | |
| $\Delta R^2$ | 0.008 | | |

Note. $R^2$ and $\Delta R^2$ values are from the first imputation data set, as SPSS does not provide $R^2$ and $\Delta R^2$ values for the pooled data sets.

determine its impact on (a) student leadership skills (primary outcome) and leadership self-efficacy (i.e., confidence to lead; secondary outcome), as well as (b) components of physical literacy (secondary outcomes) of younger students who received the transformational leadership informed FMS development intervention. Utilization of a peer-leader delivery system provided an opportunity for leadership skills to be taught and practiced. The PLPL trial utilized a train-the-trainer implementation strategy (the intervention delivered by teachers) that we believed was potentially scalable within the Canadian elementary school context.

Ultimately, we did not find evidence that the primary outcome, teacher rated transformational leadership of Grade 6/7 students could be changed by a seven-session leadership learning program and the subsequent implementation of a 10-week peer-led program. Further, there was no additional evidence of intervention effects for secondary outcomes, components of physical literacy in terms of increased FMS, self-perceptions, perceived competence, motivation, or physical activity in younger Grade 3/4 students who participated in the 10-week program. These results contradict our earlier pilot study [14].

'Although this was an adapted GLASS program, we believe there are two critical factors that may explain our null findings in the current efficacy trial. First, the delivery mechanism for the training of peer leaders was modified. The intention with this different delivery method was to test a more realistic, scalable approach that mimicked ways in which teachers receive other professional learning opportunities and continuing education credits. In this trial, the research team led a workshop with teachers and those teachers subsequently delivered the content to their

**Table 5. Grade 3/4 FMS and physical activity outcomes.**

**Maximum Throw Speed**

| Fixed Effects | Parameter Estimates | Standard Error | p-value |
|---|---|---|---|
| Intercept | 4.088 | 0.707 | < .001 |
| Baseline | 0.726 | 0.050 | < .001 |
| Gender | 1.414 | 0.344 | < .001 |
| Condition | 0.449 | 0.294 | .128 |
| $R^2$ | 0.770 | | |
| $\Delta R^2$ | 0.007 | | |

**Maximum Kick Speed**

| Fixed Effects | Parameter Estimates | Standard Error | p-value |
|---|---|---|---|
| Intercept | 6.084 | 1.451 | < .001 |
| Baseline | 0.706 | 0.086 | < .001 |
| Gender | 0.301 | 0.415 | .467 |
| Condition | 0.516 | 0.667 | .440 |
| *Variance Reduction* | | | |
| Level 1 | .000 | | |
| Level 2 | .055 | | |

**Throw-Catch Combination Score**

| Fixed Effects | Parameter Estimates | Standard Error | p-value |
|---|---|---|---|
| Intercept | 3.891 | 0.497 | < .001 |
| Baseline | 0.463 | 0.066 | < .001 |
| Gender | 1.123 | 0.432 | < .01 |
| Condition | 0.068 | 0.419 | .870 |
| $R^2$ | 0.263 | | |
| $\Delta R^2$ | 0.000 | | |

**Throw Process Scores (as assessed by the Test of Gross Motor Development-3$^{rd}$ edition)**

| Fixed Effects | Parameter Estimates | Standard Error | p-value |
|---|---|---|---|
| Intercept | 1.325 | 0.272 | < .001 |
| Baseline | 0.452 | 0.071 | < .001 |
| Gender | 1.985 | 0.329 | < .001 |
| Condition | 0.054 | 0.266 | .839 |
| $R^2$ | 0.497 | | |
| $\Delta R^2$ | 0.000 | | |

**Kick Process Scores (as assessed by the Test of Gross Motor Development-3$^{rd}$ edition)**

| Fixed Effects | Parameter Estimates | Standard Error | p-value |
|---|---|---|---|
| Intercept | 2.513 | 0.417 | < .001 |
| Baseline | 0.269 | 0.055 | < .001 |
| Gender | 0.246 | 0.252 | .330 |
| Condition | -0.733 | 0.445 | .099 |
| *Variance Reduction* | | | |
| Level 1 | -0.002 | | |
| Level 2 | 0.238 | | |

**Moderate to Vigorous Physical Activity During the School Day**

| Fixed Effects | Parameter Estimates | Standard Error | p-value |
|---|---|---|---|
| Intercept | 14.036 | 4.447 | < .01 |
| Baseline | 0.569 | 0.136 | < .001 |
| Gender | 8.991 | 3.042 | < .01 |
| Condition | -3.841 | 3.033 | 0.212 |

*(Continued)*

**Table 5.** (Continued)

| | | | |
|---|---|---|---|
| $R^2$ | 0.435 | | |
| $\Delta R^2$ | 0.020 | | |

Note. $R^2$ and $\Delta R^2$ values are from the first imputation data set, as SPSS does not provide $R^2$ and $\Delta R^2$ values for the pooled data sets.

Grade 6/7 students. The older students then delivered the physical literacy program to younger students in their school. In contrast, our pilot study focused on peer leaders allowed for the research team to travel to the one intervention school and directly work with the students. The current study therefore had an additional messenger (i.e., the teacher) to relay information from the research team to the students. Balanced against the null findings for primary and secondary outcomes, the current trial results suggest that some critical information or quality control may have been lost in translation with teachers acting as the mediating party between the research team and the students. While teachers self-reported that all lessons were provided to their Grade 6/7 classes, we did not conduct a process evaluation for this aspect of the intervention.

A second potential reason for null effects of this intervention may be due to a decrease in intervention effectiveness that has been shown to be a common occurrence when moving to larger-scale trials [53]. The change from a highly successful pilot study in a small number of schools leading to a randomized controlled trial with null findings would suggest that the current study's different delivery modality did not work as anticipated [54, 55]. It could also be that by replicating and extending this work, that in hindsight, the GLASS study could have been flawed. As both studies used many of the same strategies, we would expect similar results, but this was not the case. Possible threats to external validity, such as the larger number of schools and/or the change in age of the younger participants could also explain possible differences in results of the current trial. Our pilot study only included one intervention school, it was much easier for the research team to provide implementation support. In the current trial, the larger number of schools (3 intervention schools) did not allow for the research team to provide on-going support. Members of the research team viewed lessons delivered by the Grade 6/7 students early in the 10-week program and provided feedback to the leaders. However, these observations only occurred in Weeks 1–3 of the FMS programming, not across the full ten weeks. Process evaluation information would be critical to possibly understand where there may have been a diversion from planned activities when researchers were not present. It is notable that in terms of fidelity of delivering the physical literacy component of the intervention, only one session at one school was missed. This information was self-reported by the teachers to the study coordinator.

From a measurement perspective, failure to identify a statistically significant effect for the primary outcome could be a result of ceiling effects with the primary outcome, which was scored on a five-point scale [33]. Baseline levels of teacher rated transformational leadership were already quite high. Thus, even if teacher ratings of their students changed (i.e., increased over time) there may have been minimal room for improvement. For example, intervention school teachers had a mean transformational leadership rating of 2.96 at baseline. This score increased to 3.25 at post-test, but the max score on this scale was a 4.0. Future trials that decide to use a similar scale may consider including a specific inclusion criterion relating to low scores on transformational leadership to ensure there is sufficient reason to intervene. Related to the movement skills of the younger children participating in this study, the dose may not have been sufficient to induce change. Notably, lessons across the ten-week program focused on six different object control skills each of which were the focus of a lesson three (catch, throw, two-handed

strike, dribble) or four (kick, underhand throw) times across the 20 possible sessions. Thus, only half of the intervention time for younger students was spent learning and developing the skills that were assessed. It is plausible that students need additional time, practice and feedback to improve these skills in a clinically meaningful way. It could also be that younger children (Kindergarden, Grade 1, Grade 2) have less developed skills and thus are more likely to demonstrate skill development after participation in an intervention. The current study had Grade 3 and 4 students, which while only a few years older, meant their skills may have been more developed and less susceptible to change with a similar dose of intervention as given to younger students. As it relates to movement skills there may be a critical window of opportunity for interventions to see the largest improvements. The same could be possible for the psychological constructs that were assessed; however, longitudinal studies or long-term intervention follow-up would be needed to confirm this 'critical window' hypothesis.

Existing evidence related to the use of a train-the-trainer model to deliver the intervention could be described as mixed [6, 56]. Thus, while the current study has null findings, this is not out of the ordinary. Indeed, a recent review of peer-led interventions found that while physical activity can be improved through such interventions. For the vast majority of behavioral, psychological, physiological, and leadership outcomes there were limited significant improvements [6]. In contrast to these results, other intervention delivery modalities that use teachers or other trained facilitators appear to be more successful. Lander and colleagues [56] report that providing training to teachers, who subsequently lead and deliver school-based interventions, have shown improvements in physical activity behaviors and improved motor skills. Further still, programs such as 'Supporting Children's Outcomes using Rewards, Exercise and Skills' (SCORES) intervention [15] have shown that trained facilitators who receive ongoing support can lead to improvements in physical activity, movement skills, and other health-related outcomes. Yet, when scaled-up and less on-going support is provided, effect sizes were much smaller [57]. Train-the-trainer strategies are important and necessary as they increase the feasibility of delivering programs. But who is trained and how much training they receive also plays a vital role for potential outcomes of the interventions.

Strengths of the current study include the randomized design, the use of a train-the trainer implementation strategy which mimics a more realistic strategy than an intervention run exclusively by researchers, and the collection of a variety of measures at multiple levels (teacher, leaders, younger students). There are also limitations in the current study including the smaller sample than originally planned, which decreased our statistical power. As highlighted earlier, the study sample size was affected due to the mandatory pause in the study due to the COVID-19 pandemic. Our initial power analyses necessitated that a total of 14 schools, 336 peer leaders, and 1,008 younger students would be needed to detect a medium effect size. In reality, we recruited 6 schools, 132 peer leaders, and 227 younger students. So we were underpowered and thus conclusions should be interpreted with caution. Other limitations include the lack of a detailed process evaluation across the duration of the intervention, which limits our ability to understand the true dose of intervention received by all participants. Lastly, we did not assess movement skill competence of the peer-leaders and as such cannot make determinations as to whether the differences in skill capability of these instructors contributed to null-findings related to changes in fundamental movement skills.

## Conclusion

Based upon prior successful work, the evaluation of our leadership and physical literacy intervention in a larger scale efficacy study was warranted. Ultimately, no significant findings were found, suggesting that modifications to the current design are required if changes in primary

and secondary outcomes are desired. Peer-led interventions still represent a promising, feasible modality for implementing such interventions [6]. To improve the likelihood of impact, greater attention on implementation strategies, which address schools' adoption of such peer leadership programs, should be considered as part of any future trials.

## Supporting information

**S1 Fig. SPIRIT diagram.**
(DOCX)

**S1 Table. TIDieR (Template for Intervention Description and Replication) checklist.**
(DOCX)

**S2 Table. Intervention lessons, goals, and exemplar activities.**
(DOCX)

**S3 Table. Teacher and grade 6/7 student outcomes Cronbach alpha, ICC, and design effects.**
(DOCX)

**S4 Table. Grade 3/4 student outcomes Cronbach alpha, ICC, and design effects.**
(DOCX)

**S5 Table. Peer leader program evaluation.**
(DOCX)

**S6 Table. CONSORT statement for cluster randomized controlled trials.**
(DOCX)

**S1 File. Surveys and questionnaires.**
(DOCX)

**S2 File. Clinical trials registration.**
(PDF)

**S3 File. Approved study protocol.**
(PDF)

**S4 File. Deidentified leader dataset.**
(SAV)

**S5 File. Deidentified young student dataset.**
(SAV)

## Acknowledgments

The authors would like to thank all participating students, teachers, and schools. We also thank all of the undergraduate research assistants who helped with this project.

## Author Contributions

**Conceptualization:** Ryan M. Hulteen, David R. Lubans, Ryan E. Rhodes, Guy Faulkner, Yan Liu, Patti-Jean Naylor, Nicole Nathan, Katrina J. Waldhauser, Colin M. Wierts, Mark R. Beauchamp.

**Data curation:** Mark R. Beauchamp.

**Formal analysis:** Ryan M. Hulteen, Yan Liu, Colin M. Wierts, Mark R. Beauchamp.

**Funding acquisition:** Mark R. Beauchamp.

**Investigation:** Ryan M. Hulteen, Mark R. Beauchamp.

**Methodology:** Ryan M. Hulteen, Mark R. Beauchamp.

**Project administration:** Ryan M. Hulteen, Mark R. Beauchamp.

**Resources:** Mark R. Beauchamp.

**Supervision:** Mark R. Beauchamp.

**Writing – original draft:** Ryan M. Hulteen, David R. Lubans, Ryan E. Rhodes, Guy Faulkner, Yan Liu, Patti-Jean Naylor, Nicole Nathan, Katrina J. Waldhauser, Colin M. Wierts, Mark R. Beauchamp.

**Writing – review & editing:** Ryan M. Hulteen, David R. Lubans, Ryan E. Rhodes, Guy Faulkner, Yan Liu, Patti-Jean Naylor, Nicole Nathan, Katrina J. Waldhauser, Colin M. Wierts, Mark R. Beauchamp.

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
