## [Decision Letter · Decision Letter 0]

28 Sep 2022

PONE-D-22-11797Evaluation of the Peer Leadership for Physical Literacy Intervention: A Cluster Randomized Controlled TrialPLOS ONE

Dear Dr. Huteen,

Thank you for submitting your manuscript to PLOS ONE. After careful consideration, we feel that it has merit but does not fully meet PLOS ONE’s publication criteria as it currently stands. Therefore, we invite you to submit a revised version of the manuscript that addresses the points raised during the review process.

We look forward to receiving your revised manuscript.

Kind regards,

Venkat Rao Vishnumolakala

Academic Editor

PLOS ONE

2.Thank you for stating the following in the Funding Section of your manuscript:

“Funding for this project is provided by the Social Sciences and Humanities Research Council of Canada (SSHRC) [Insights Grant #435-2017-0268]. The funding body (SSHRC) had no role in the design, data collection, analysis, interpretation of data, or the writing of the manuscript. DRL is supported by a National Health and Medical Research Council Senior Research Fellowship (APP1154507).”

“Author M.B. received funding to conduct this project through the Social Sciences and Humanities Research Council of Canada (SSHRC; https://www.sshrc-crsh.gc.ca/home-accueil-eng.aspx) Insights Grant #435-2017-0268. Author DRL is supported by a National Health and Medical Research Council Senior Research Fellowship (APP1154507; https://www.nhmrc.gov.au/). The funding bodies, in both instances, had no role in the design, data collection, analysis, interpretation of data, or the writing of the manuscript.”

3.We note that you have indicated that data from this study are available upon request. PLOS only allows data to be available upon request if there are legal or ethical restrictions on sharing data publicly. For more information on unacceptable data access restrictions, please see http://journals.plos.org/plosone/s/data-availability#loc-unacceptable-data-access-restrictions.

4.We note that you have stated that you will provide repository information for your data at acceptance. Should your manuscript be accepted for publication, we will hold it until you provide the relevant accession numbers or DOIs necessary to access your data. If you wish to make changes to your Data Availability statement, please describe these changes in your cover letter and we will update your Data Availability statement to reflect the information you provide.

Additional Editor Comments:

Dear Dr. Huteen,

Thank you for submitting your paper to PLOS One; and patiently waiting for the outcome of the peer review process.

Please address the reviewer's concerns and submit your revised manuscript in accordance with PLOS-One's author guidelines.

Kind Regards,

Ven

Reviewers' comments:

Reviewer's Responses to Questions

**Comments to the Author**

1. Is the manuscript technically sound, and do the data support the conclusions?

Reviewer #1: Partly

2. Has the statistical analysis been performed appropriately and rigorously? 

Reviewer #1: Yes

3. Have the authors made all data underlying the findings in their manuscript fully available?

Reviewer #1: Yes

4. Is the manuscript presented in an intelligible fashion and written in standard English?

Reviewer #1: Yes

5. Review Comments to the Author

Reviewer #1: Dear Editors and Authors,

Thank you for your patience in my review of this manuscript.

I have carefully read the manuscript titled, “Evaluation of the Peer Leadership for Physical Literacy Intervention: A Cluster Randomized Control Trial”. Overall the manuscript is well-written, and the intervention itself appears to be novel. The train-the-trainer approach is particularly exciting and, as the authors pointed out, is a key component missing from physical literacy interventions. Ultimately, the results of the paper are underwhelming and could be interpreted as disappointing from a statistical significance perspective; however, I strongly believe there is much knowledge to be gained from well-conducted research, regardless of final p-values.

Even so, I have several comments that I believe need to be addressed before the paper is acceptable for publication. I hope the authors find these comments helpful as they continue their work with this manuscript. These comments are primarily arranged chronologically and not necessarily in order of scale.

1. The abstract is confusing. The purpose statement (listed as background) does not include an outcome variable, so the reader has no context regarding the purpose of the study. Secondly, the study timeline is unclear. L50-51 read as if there was a follow-up assessment and not an immediate post-intervention assessment.

2. Why is a teacher-reported outcome the primary outcome? If the teachers were also the ones leading the training/intervention, as it appears to be, is there not a conflict of interest regarding their ability to score objectively? Also, what is the rationale behind having a teacher- and student-reported scores of transformational leadership?

3. Physical literacy is never clearly defined in the introduction (only L110). To have a better justification for the multiple outcomes, the introduction needs to have a well-articulated definition of physical literacy and the need for programs to enhance multiple aspects of physical literacy.

4. The GLASS study is first introduced as an external study (L100) but later reported as a pilot study (L210). Greater transparency in the relationship between the GLASS and PLPL intervention is needed in the introduction.

5. L116 mentions how the GLASS intervention influenced PA, but this outcome is not listed above.

6. Please revise the manuscript for clarity regarding the multiple outcomes. Sometimes they are presented as primary, secondary, or tertiary (results section), whereas other times they are simply listed as peer-leader leadership skills and student physical literacy outcomes (example L138-140). If a greater definition of physical literacy was provided in the introduction, I would suggest presenting as leadership skills and physical literacy outcomes. It is unclear why physical activity, which is an important component of physical literacy, is considered tertiary compared to other PL factors.

7. Is there a justification for the two modifications made from GLASS to PLPL (L212)?

8. L219- Were teachers compensated for their time?

9. It is unclear how the intervention delivered from the peer-leaders to students was a physical literacy program. The authors mention that this intervention included “warm-up, demonstrate the movement skill, skill practice, and set-up and oversee the game” (L277-278), but this appears like the intervention was only targeting one aspect of physical literacy, movement competence (note: L261- it is referred to as a movement skills program). If indeed this was a movement skills intervention, the authors provide little rationale to include other physical literacy outcomes in the results such as perceived competence and physical activity.

10. Was there a theoretical approach used in the physical literacy intervention?

11. Why were only object control skills targeted?

12. Was the movement competence of peer-leaders assessed? If not, how are you sure they have the competence to lead skill instruction?

13. L283- what does “during class time” refer to here?

14. L354-355: Did you modify the scale? If so, what modifications were made?

15. L374- The overall description of motor assessment is lacking. How were motor skills assessed? Did you follow TGMD-3 instructions for assessment (e.g., skill demonstration, practice trial, test trials)? Did you do a different assessment approach? What is the throw/catch assessment? Was throwing scored during this assessment or just the number of catches? Was there a time limit for this test?

16. L393-Since the catching task is not fully described, I have no way to interpret this statement.

17. Please provide greater IRR details on raters who scored the TGMD test trials.

18. L395- was the physical activity data collected using a stratified random sampling technique to ensure equal number of girls and boys? It does not appear so, but I think this would be important based on established sex-difference in this outcome.

19. L491- Wouldn’t sex be a more appropriate term here?

20. L528- Please double-check for consistent language- fundamental movement skills, motor skills, motor competence, and motor skill outcomes are all used throughout this manuscript.

21. L535- if the interaction was non-significant, why is it re-included in the model and then interpreted (note: this appears to be further included as a finding in the discussion L629-631).

22. Table 5- Throwing and kicking speed are well defined in the methods, but throw-catch combination score, throw components, and kick components are not clear. I am assuming that “components” is synonymous with TGMD or process. If that is the case, please revise the language in table 5 to align with the text.

23. L576- Please confirm that five-session is correct. L255 mentions 7.

24. It would be nice to have some greater depth in the discussion relating to other literature. For example, how do the null findings relate to other literature using similar train-the-trainer approaches (L592-596)? Is it that the pilot study was flawed (L604) of that it was implemented at a more critical window of motor skill development and malleability? What longitudinal data is available on motor skills and psychological constructs measured (L649)?

25. L613- What is FMS?

6. PLOS authors have the option to publish the peer review history of their article (what does this mean?). If published, this will include your full peer review and any attached files.

Reviewer #1: No

---

## [Author Response · Author response to Decision Letter 0]

1 Nov 2022

PONE-D-22-11797

Evaluation of the Peer Leadership for Physical Literacy Intervention: A Cluster Randomized Controlled Trial

Reviewer’s Comments in bold

Authorship Team Response in normal text

Revised text within the manuscript in italics

Thank you for your patience in my review of this manuscript.

I have carefully read the manuscript titled, “Evaluation of the Peer Leadership for Physical Literacy Intervention: A Cluster Randomized Control Trial”. Overall the manuscript is well-written, and the intervention itself appears to be novel. The train-the-trainer approach is particularly exciting and, as the authors pointed out, is a key component missing from physical literacy interventions. Ultimately, the results of the paper are underwhelming and could be interpreted as disappointing from a statistical significance perspective; however, I strongly believe there is much knowledge to be gained from well-conducted research, regardless of final p-values.

Even so, I have several comments that I believe need to be addressed before the paper is acceptable for publication. I hope the authors find these comments helpful as they continue their work with this manuscript. These comments are primarily arranged chronologically and not necessarily in order of scale.

We would like to thank the Reviewer for their thorough review and helpful comments. We are glad that they found the manuscript to be relevant and interesting despite the lack of statistical significance of our findings. We have provided our responses to the Reviewers’ suggestions for improvement. We believe the manuscript has been strengthened and hope that it is now in acceptable form for publication in PLOS ONE.

1. The abstract is confusing. The purpose statement (listed as background) does not include an outcome variable, so the reader has no context regarding the purpose of the study. Secondly, the study timeline is unclear. L50-51 read as if there was a follow-up assessment and not an immediate post-intervention assessment.

We have amended the abstract to address the Reviewer’s concerns. Specific changes that were made to the abstract include: 1) changing the first heading from “Background” to “Purpose”, 2) we have now moved the listing of the primary and secondary outcomes to the Purpose section of the abstract. This was done to provide further context about the focus and emphasis of the intervention, and 3) we changed our language to better articulate that the follow-up assessment was done immediately post-intervention. We have provided the first two sections of the abstract with these changes below for convenience. These changes can also be viewed on lines 37-53.

Purpose: The purpose of this research was to develop, implement, and test the efficacy of a theory-driven, evidence-informed peer leadership program for elementary school students (Grade 6 and 7; age 11-12 years) and the Grade 3/4 students with whom they were partnered. The primary outcome was teacher ratings of their Grade 6/7 students’ transformational leadership behaviors. Secondary outcomes included: Grade 6/7 students’ leadership self-efficacy, as well as Grade 3/4 student motivation, perceived competence, general self-concept, fundamental movement skills, school-day physical activity, program adherence, and program evaluation. 

Methods: We conducted a two-arm cluster randomized controlled trial. In 2019, 6 schools comprising 7 teachers, 132 leaders, and 227 grade 3 and 4 students were randomly allocated to the intervention or waitlist control conditions. Intervention teachers took part in a half-day workshop (January 2019), delivered 7 x 40 minute lessons to Grade 6/7 peer leaders (February and March 2019), and these peer leaders subsequently ran a ten-week physical literacy development program for Grade 3/4 students (2x30 minute sessions per week). Waitlist-control students followed their usual routines. Assessments were conducted at baseline (January 2019) and immediately post-intervention (June 2019). 

2. Why is a teacher-reported outcome the primary outcome? If the teachers were also the ones leading the training/intervention, as it appears to be, is there not a conflict of interest regarding their ability to score objectively? Also, what is the rationale behind having a teacher- and student-reported scores of transformational leadership?

There are a couple of different reasons why teacher ratings of Grade 6/7 students was chosen as the primary outcome for this study. First, transformational leadership is the most proximal outcome related to the intervention that was delivered (that is, that construct was the primary target as part of the intervention, with all other outcomes putatively deriving from that). Had we chosen student self-reported ratings of transformational leadership there may have been a higher likelihood of an inflated, self-serving bias within peer-leaders’ own ratings. Teachers on the other hand are one-step removed, but since they interact and observe these students on an on-going basis while at school, the teacher was seen as the best person to report on leadership behaviors. Indeed, teachers are regularly required to assess students’ competency in a range of academic outcomes. No other third party in a school setting would be as well equipped to rate an entire class of students, as they would be scoring without any context, thus being potentially subject to Hawthorn effects. Similarly, had we used an external rater (e.g. researcher) to appraise students’ behaviors, such assessments (e.g., over 2-3 classes) would provide a very limited timeframe to inform those appraisals. In addition, due to the use of an experimental design, if any inflation in teacher ratings occurred, due to the use of random allocation, one would expect that any potentially inflated scores would be equivalent across conditions. 

Although we operationalized teacher-ratings of students’ peer leadership as our primary outcome, we also included students’ self-ratings as a secondary outcome. For the reasons/limitations we highlight above it was not considered as robust a measure as teacher ratings of peer-leaders’ behaviours. Had there been differences in intervention effects for self- versus teacher-rated measures, we saw potential to examine (as an exploratory question) discrepancies between these two ratings and whether those discrepancies accounted for some of the study findings (e.g., potentially elevated peer-leader self-efficacy). As the intervention effects for both self- and teacher-ratings were null, we feel that any discussion of why we used self-ratings of leadership is somewhat redundant. However, for the sake of full transparency (and in alignment with our pre-trial registration) we believe that data derived from both sets of measures should be reported. 

3. Physical literacy is never clearly defined in the introduction (only L110). To have a better justification for the multiple outcomes, the introduction needs to have a well-articulated definition of physical literacy and the need for programs to enhance multiple aspects of physical literacy.

We have expanded our Introduction to clearly articulate the definition of physical literacy. It is now defined in the text on lines 110-112. 

Physical literacy is defined as the “motivation, confidence, physical competence, knowledge, and understanding to value and take responsibility for engagement in physical activities for life” (20). 

Additionally, we have provided rationale for why interventions that target components of physical literacy are important. As supported by a recent publication by Carl et al., (2022) physical literacy “cultivates a holistic and integrative understanding of human movement” as it allows us to target multiple outcomes at once and intervention evidence suggests that a composite measure of physical literacy can indeed be improved. Yet, the extent to which components of physical literacy are improved are more nuanced with physical components easier to improve compared to psychological components. Expansion of this idea is provided below and on lines 110-124 in the manuscript.

Physical literacy is defined as the “motivation, confidence, physical competence, knowledge, and understanding to value and take responsibility for engagement in physical activities for life” (20). Physical literacy thus takes a more holistic perspective in understanding the attributes that contribute to one’s capability to be physically active across the lifespan. Multiple interventions have focused on individual components of physical literacy from a physical, psychological, or social perspective (21). However, targeting multiple components of physical literacy within a single intervention may be more advantageous, as it will help elucidate those factor(s) that best assist an individual in being physically active. For example, a recent systematic review and meta-analysis of physical literacy interventions demonstrated that there were significant treatment effects for interventions when all physical literacy outcomes were combined (21). Yet, these results are more nuanced with interventions exerting differential effects on the varying outcome categories. This same review describes that there appears to be the strongest evidence for improving physical competence, while psychological outcomes (e.g., motivation, confidence) are harder to improve through interventions (21). 

Citation: Carl, J., Barratt, J., Wanner, P. et al. The effectiveness of physical literacy interventions: A systematic review with meta-analysis. Sports Medicine (2022). https://doi.org/10.1007/s40279-022-01738-4

4. The GLASS study is first introduced as an external study (L100) but later reported as a pilot study (L210). Greater transparency in the relationship between the GLASS and PLPL intervention is needed in the introduction.

We agree with the Reviewer that the language previously used was confusing. We have amended the document throughout to clearly articulate that the GLASS trial is a pilot study. We have highlighted key modifications from the pilot GLASS study to the current PLPL trial on lines 227to 238. This text is also provided below for convenience. 

Two key changes were made to the current trial compared to our earlier GLASS trial. First, the target age group of younger students was different, shifting from Kindergarten, Grade 1 and 2 in our previous non-randomized trial to Grade 3 and 4 students in this efficacy trial. Secondly, previously where a research team delivered all content to peer leaders, a train-the-trainer approach was adopted and the research team delivered all content to Grade 6/7 teachers through a half-day workshop at the last author’s university. These teachers then delivered the content to their students. This change in delivery mechanism was designed with intervention scalability and sustainability foregrounded. Specifically, having teachers deliver programs within their own classes are more likely to be sustained and delivered at scale compared to an external research team that would be required to deliver a program across multiple schools on an ongoing basis. 

While the current PLPL has a few changes compared to the GLASS pilot study, according to a recent review by Beets et al., 2020, it is common for changes to be made when scaling-up interventions. 

Beets, M., Weaver, R. G., Ioannidis, J. P. A., Geraci, M., Brazendale, K., Decker, L., . . . Milat, A. J. (2020). Identification and evaluation of risk of generalizability biases in pilot versus efficacy/effectiveness trials: a systematic review and meta-analysis. International Journal of Behavioral Nutrition and Physical Activity, 17(1), 19. doi:10.1186/s12966-020-0918-y

5. L116 mentions how the GLASS intervention influenced PA, but this outcome is not listed above.

We have reported that physical activity was assessed in the initial GLASS study (line 101). However, to be more clear, we have included reference to a specific non-significant finding related to school-day physical activity at the end of the paragraph for transparency about findings in the GLASS study. This physical activity result can be found on line 128-130. 

The study did not result in a statistically significant difference in terms of school day physical activity (d = 0.29; p = 0.313).

6. Please revise the manuscript for clarity regarding the multiple outcomes. Sometimes they are presented as primary, secondary, or tertiary (results section), whereas other times they are simply listed as peer-leader leadership skills and student physical literacy outcomes (example L138-140). If a greater definition of physical literacy was provided in the introduction, I would suggest presenting as leadership skills and physical literacy outcomes. It is unclear why physical activity, which is an important component of physical literacy, is considered tertiary compared to other PL factors.

To best align with our pre-registration of this trial on clinicaltrials.gov, we have sought to use consistent language in our manuscript as that used on this pre-registration website. As such, we specifically use the terms primary outcome and secondary outcomes. To be clear, the primary outcome of this trial is teacher ratings of Grade 6/7 students’ transformational leadership behaviors. All other outcomes are secondary outcomes in this trial. These secondary outcomes include both outcomes for Grade 6/7 students and Grade 3/4 students. These changes have been made throughout the manuscript. Additionally, after reviewing our preregistration where we listed physical activity as a secondary outcome, we have now made sure to also list physical activity as a secondary outcome throughout the manuscript.

7. Is there a justification for the two modifications made from GLASS to PLPL (L212)?

The two key modifications made to the PLPL trial compared to the GLASS trial was the target age group for the physical literacy component of the intervention and the delivery mechanism of the intervention materials. We have provided the modifications on lines 227-238 of the manuscript. The primary emphasis related to the delivery mechanism was focusing on scalability and sustainability. Ultimately change in delivery agent is the biggest contributing factor to a drop in effect size when scaling up interventions (Beets et al., 2020). If an intervention with a different delivery agent in a scaled up version is successful this points to something that we can provide to policy makers as an easily modifiable program. Relative to the change in age group, our thinking at the time of development was focused on trying to obtain valid and reliable results for the self-report measures. Further, given the success of the GLASS trial, we were interested if similar results could be found for an older age group. 

Two key changes were made to the current trial compared to our earlier GLASS trial. First, the target age group of younger students was different, shifting from Kindergarten, Grade 1 and 2 in our previous non-randomized trial to Grade 3 and 4 students in this efficacy trial. Secondly, previously where a research team delivered all content to peer leaders, a train-the-trainer approach was adopted and the research team delivered all content to Grade 6/7 teachers through a half-day workshop at the last author’s university. These teachers then delivered the content to their students. This change in delivery mechanism was designed with intervention scalability and sustainability foregrounded. Specifically, having teachers deliver programs within their own classes are more likely to be sustained and delivered at scale compared to an external research team that would be required to deliver a program across multiple schools on an ongoing basis.

Citation: Beets, M., Weaver, R. G., Ioannidis, J. P. A., Geraci, M., Brazendale, K., Decker, L., . . . Milat, A. J. (2020). Identification and evaluation of risk of generalizability biases in pilot versus efficacy/effectiveness trials: a systematic review and meta-analysis. International Journal of Behavioral Nutrition and Physical Activity, 17(1), 19. doi:10.1186/s12966-020-0918-y

8. L219- Were teachers compensated for their time?

Students received compensation for their participation in the study (depending on the measures completed) and the schools involved in the study were provided with equipment packs valued at approximately $250 CDN. The teachers themselves were not directly compensated (monetarily), but they were provided with all of the materials and resources free of charge that were required to complete the intervention (see lines 261 to 264). 

9. It is unclear how the intervention delivered from the peer-leaders to students was a physical literacy program. The authors mention that this intervention included “warm-up, demonstrate the movement skill, skill practice, and set-up and oversee the game” (L277-278), but this appears like the intervention was only targeting one aspect of physical literacy, movement competence (note: L261- it is referred to as a movement skills program). If indeed this was a movement skills intervention, the authors provide little rationale to include other physical literacy outcomes in the results such as perceived competence and physical activity.

We have defined physical literacy on lines 110 to 112 as the “motivation, confidence, physical competence, knowledge, and understanding to value and take responsibility for engagement in physical activities for life”. Notably, the ‘gold standard’ definition of physical literacy is still being debated (see Carl et al., 2022 or Lounsbery and McKenzie, 2015). Until there is consensus, it is not possible to definitively conclude what does or does not need to be assessed to be considered a “physical literacy” intervention. There is clear alignment between the definition of physical literacy and secondary outcomes related to the Grade 3/4 students, such as fundamental movement skills (skill), perceived competence (attitude), motivation (attitude), self-concept (attitude), and physical activity. While we did not have an explicit measure of “knowledge” in this particular intervention, there is clearly a knowledge component to our intervention, as the peer leaders used key teaching points when instructing when trying to develop the movement skills of these younger students. Additionally, Peer Leaders were taught strategies to motivate their students based on Transformational Leadership theory (e.g., being an enthusiastic leader). Peer leaders also learnt to deliver activity sessions that were designed to improve students' perceived and actual motor competence (developmentally appropriate activities should have effects on both perceived and actual competence). Finally, peer leaders were encouraged to remind students about the importance of learning movement skills and how they transfer to different sports and activities. This could be considered “knowledge”. 

Citations: 

Carl J, Barratt J, Wanner P, Töpfer C, Cairney J, Pfeifer K. The Effectiveness of Physical Literacy Interventions: A Systematic Review with Meta-Analysis. Sports Med (2022). https://doi.org/10.1007/s40279-022-01738-4

Lounsbery, M.A.F. and McKenzie, T.L. Physically Literate and Physically Educated: A Rose By Any Other Name? J Sports Health Sci (2015). Doi: 10.1016/j.jshs.2015.02.002

In fact, when we look at other intervention studies included in a systematic review of physical literacy interventions (Carl et al., 2022) we can see multiple examples of studies where the central component of the intervention was focused on the movement skill instruction. Further, when examining these interventions, they too did not include explicit sessions focused solely on each component of physical literacy. They also did not measure all components of physical literacy. Two examples include the following studies: 

Bremer, E., Graham, J.D., and Cairney, J. (2020). Outcomes and feasibility of a 12-week physical literacy intervention for children in an afterschool program. International Journal of Environmental Research and Public Health, 17, 3129. Doi: 10.3390/ijerph17093129

Pullen, B.J., Oliver, J.L., Lloyd, R.S., and Knight, C.J. (2020). The effects of strength and conditioning in physical education on athletic motor skill competencies and psychological attributes of secondary school children: a pilot study. Sports, 8, 138. Doi: 10.3390/sports8100138

10. Was there a theoretical approach used in the physical literacy intervention?

Consistent with our pilot GLASS study, this intervention was informed by Transformational Leadership Theory as described in “Peer-Leadership Development Program (Intervention Part 1)”. See lines 248-260. This was done because the primary outcome of the intervention was teacher-ratings of Grade 6/7 transformational leadership. Core components of our intervention framework that was informed by Transformational Leadership Theory (as per Kelloway and Barling, 2000) include a) providing opportunities to practice those behaviors and b) receiving feedback on the implementation of those strategies. Both of these components took place during the physical literacy intervention as the direct instruction from Grade 6/7 students to Grade 3/4 students represents practicing leadership behaviors. Receiving feedback took place in the first couple of weeks of the physical literacy component of the intervention as members of the research team provided direct feedback to Grade 6/7 students after watching their lessons. There was no separate theoretical approach focused specifically related to physical literacy, as this was not the primary outcome of the intervention. 

Citations: 

Kelloway EK and Barling J. What we have learned about developing transformational leaders. Leadership Org Dev J. 2000; 21(1): 355-362. 

11. Why were only object control skills targeted?

Object control skills were targeted for three key reasons. First, the initial GLASS study only focused on object control skills. Given the large effect size associated with an improvement of object control skills in the GLASS trial, we were justified in once again targeting these skills. Second, existing evidence in the field of motor development repeatedly demonstrates that object control skills are a better predictor of current and future physical activity participation compared to locomotor skills (Barnett et al., 2009; Pienaar et al., 2021). Third, the choice in how many skills to assess was in part made due to pragmatic and feasibility reasons. Working in schools is a significant undertaking and often involves a condensed period of time to collect data. In the case of this study, we were constrained to assessing 20-30 students in a one-hour period, thus it was not feasible to assess all six object control skills that aligned with lessons nor expand beyond object control skills and include locomotor and stability skills. 

Citations: 

Barnett, L. M., Beurden, E. van, Morgan, P. J., Brooks, L. O. & Beard, J. R. Childhood motor skill proficiency as a predictor of adolescent physical activity. J Adolesc Health 44, 252–259 (2009).

Pienaar AE, Gericke C, Plessis WD. Competency in Object Control Skills at an Early Age Benefit Future Movement Application: Longitudinal Data from the NW-CHILD Study. Int J Environ Res Public Health. 2021 Feb 9;18(4):1648

12. Was the movement competence of peer-leaders assessed? If not, how are you sure they have the competence to lead skill instruction?

The movement competence of peer-leaders was not assessed. This would certainly have been an interesting inclusion in the current study and provides an interesting future direction for this line of work. However, competence in a skill or set of skills does not necessarily equate to teaching competence either. Indeed, it may be that administering the intervention in this way mimics the increasing trend of classroom teachers being responsible for instructing physical education instead of having a specialized physical education teacher. While we may never be sure that peer-leaders have the physical competence to lead skill instruction, peer leaders were provided with a multitude of resources to facilitate their instruction including: receiving instruction from their teachers, being given practice sessions to practice their teaching, being provided with key teaching points and specific planned activities, as well as feedback from the research team during the initial weeks of the intervention. We have now included this as a limitation on lines 702 to 704.

Lastly, we did not assess movement skill competence of the peer-leaders and as such cannot make determinations as to whether the differences in skill capability of these instructors contributed to null-findings related to changes in fundamental movement skills.

13. L283- what does “during class time” refer to here?

We have deleted “during class time” so that it is not assumed that lessons were planned at the same time as the class in which lessons were being delivered. All lessons delivered by the Grade 6/7 students were developed and planned during a set time in the school day that was separate from the direct instruction periods with Grade 3/4 students.

14. L354-355: Did you modify the scale? If so, what modifications were made?

The scale was not modified. To be clear, we used the scale developed by Sebire and colleagues . This scale is provided in the Supplementary Materials for this manuscript. We provide the Sebire et al reference below if the Reviewer would like to read the specifics about this scale.

Sebire SJ, Jago R, Fox KR, Edwards MJ, and Thompson JL. Testing a self-determination theory model of children's physical activity motivation: a cross-sectional study. Int J Behav Nutr Phys Act. 2013; 10(1): 111.

15. L374- The overall description of motor assessment is lacking. How were motor skills assessed? Did you follow TGMD-3 instructions for assessment (e.g., skill demonstration, practice trial, test trials)? Did you do a different assessment approach? What is the throw/catch assessment? Was throwing scored during this assessment or just the number of catches? Was there a time limit for this test?

More details have been provided on lines 393 to 430 to specify procedures related to the motor skill assessments. To answer the Reviewer’s specific questions, motor skills were assessed using both a process and product approach. The process approach entails the observation of behavioral components of a skill and making a subjective determination as to whether the varying skill criteria were completed or not. The product approach entails quantifying the outcome of the movement. Overhand throw and kicking were assessed via a process approach using criteria from the Test of Gross Motor Development-3rd edition. Product outcomes were included for throwing, kicking, and catching. Throw and kick speed were recorded and number of successful catches were recorded. While the criteria for the Test of Gross Motor Development-3rd edition were used, there were two modifications to the traditional protocols. First, there was no demonstration of the throw and kick. This is due to the fact that all students knew what these skills were and we were interested in understanding their ‘true’ capability. Additionally, since we were interested in the product outcome, we instructed students to throw/kick the ball as hard as they could. Instructing participants in this way helps to elicit their most developmentally advanced movement pattern (Barnett et al., 2020). The other modification was that instead of completing two trials, we completed five trials of throwing and kicking. All participants were allowed a practice trial prior to their actual scored assessments. 

The throw-catch assessment is a newer assessment, which we have more explicitly explained in the manuscript. In this dual task, the only assessment was number of successful catches. Thus, the throw was not scored. This is in part due to the fact that individuals could choose to throw a ball any way they wanted (underhand, overhand, side-arm). As now stated in the manuscript, this assessment was completed twice with each trial lasting 30 seconds each to complete as many throws and catches as possible. Participants were given a practice trial as well, if they wanted.

Citation:

Barnett, L. M., Stodden, D. F., Hulteen, R. M. & Sacko, R. S. 19 Motor Competence Assessment. in The Routledge Handbook of Youth Physical Activity (eds. Brusseau, T. A., Fairclough, S. J. & Lubans, D. R.) 384–408 (2020).

FMS: Three FMS (throw, kick, catch) were assessed as a measure of object control skill competence. Process scores for kicking and throwing were assessed using criteria from the Test of Gross Motor Development-3rd edition (40). Process scores for catching were not recorded, as catching was completed in a dual-task format making it inappropriate to use criteria from the Test of Gross Motor Development-3rd edition. Scores derived from these measures have shown adequate factorial validity (i.e., one factor model for FMS competence; (χ2 (65) = 327.61, p < .001, comparative fit index = .95, Tucker-Lewis index = .94, root mean square error of approximation = .10) and test-retest reliability (intraclass correlation coefficient = 0.95-0.97) (40, 41). Participants were filmed from the side and rear-view using iPads. Skill demonstrations were not provided, but practice attempts were allowed. Two research assistants, blinded to group allocation (e.g., intervention, control) completed all coding of throwing and kicking process scores. Each research assistant was provided with one hour of training by the project coordinator (who has 10+ years experience of motor skill assessment research). This training included explanation of skill criteria and practice coding. Prior to the beginning of formalized coding, both research assistants coded 10% of all kicking and throwing videos. Research assistants had to obtain ≥ 90% agreement at the component level for both skills with the project coordinator, which was achieved. 

Product scores were also assessed for kicking (maximal speed) (42, 43), overhand throwing (maximal speed) (42, 43), and catching (number of successful catches; 44). Throw and kick speed were completed by asking participants to throw or kick a ball as hard as they could at a wall 20 feet away. Speed was measured using a Stalker Pro II radar gun. The capability to assess skills using product and process scores provides a more holistic understanding of one’s level of competence and has shown moderate to strong agreement with process scores in pediatric populations (45, 46). In accordance with previous studies using product scores and the use of the best score (42, 43, 47), we used five complete trials of both the throw and kick, respectively. The dual throw-catch task required a participant to stand behind a line (approximately three times their height) away from a wall. Upon being instructed to start the task, each participant was to take a tennis ball and throw it at a blank wall (using any throw pattern they wanted) and then try to catch the ball (on the bounce or out of the air) while remaining behind the line. Participants were free to move about the area as much as they wanted. The only place they could not go was in front of their designated line. If the thrown ball did not come back across the participant’s line or went too far away, the participant was to run to a basket of tennis balls (6 feet away from the participant) and obtain another tennis ball and continue the task. This sequence of events was repeated as many times as possible in 30 seconds with the number of catches recorded as the final result. Throwing during this task was not assessed in any way. The catching task had two trials (30 seconds each) with the best result of the two trials (most number of catches) used for the final data analyses.

16. L393-Since the catching task is not fully described, I have no way to interpret this statement.

Given the additional information provided about the throwing-catching test, we now hope that all statements related to the motor skills assessments can be better interpreted. Should the Reviewer have any further questions, we would be happy to further clarify this information. Specifics of this throw-catch tasks are now available on lines 418 to 430 and can also be read about in a previous research paper cited below. 

The dual throw-catch task required a participant to stand behind a line (approximately three times their height) away from a wall. Upon being instructed to start the task, each participant was to take a tennis ball and throw it at a blank wall (using any throw pattern they wanted) and then try to catch the ball (on the bounce or out of the air) while remaining behind the line. Participants were free to move about the area as much as they wanted. The only place they could not go was in front of their designated line. If the thrown ball did not come back across the participant’s line or went too far away, the participant was to run to a basket of tennis balls (6 feet away from the participant) and obtain another tennis ball and continue the task. This sequence of events was repeated as many times as possible in 30 seconds with the number of catches recorded as the final result. Throwing during this task was not assessed in any way. The catching task had two trials (30 seconds each) with the best result of the two trials (most number of catches) used for the final data analyses.

Terlizzi, B. et al. The relationship between functional motor competence and performance on the army combat fitness test in army reserve officer training corps cadets. Military Medicine 1–8 (2022) doi:10.1093/milmed/usab537.

17. Please provide greater IRR details on raters who scored the TGMD test trials.

Two research assistants who were blinded to group allocation were provided with one hour of training related to the assessment of throwing and kicking assessment according to the Test of Gross Motor Development-3rd edition criteria. After this training and prior to the official scoring, both raters were given 10% of all kicking and throwing videos and asked to code these. Inter-rater reliability with the project coordinator (who has 10 years of experience in the field of motor development) was calculated to be >90% for both raters. As such, raters were allowed to continuing scoring. This information can now be found on lines 402 to 409.

Two research assistants, blinded to group allocation (e.g., intervention, control) completed all coding of throwing and kicking process scores. Each research assistant was provided with one hour of training by the project coordinator (who has 10+ years experience of motor skill assessment research). This training included explanation of skill criteria and practice coding. Prior to the beginning of formalized coding, both research assistants coded 10% of all kicking and throwing videos. Research assistants had to obtain ≥ 90% agreement at the component level for both skills with the project coordinator, which was achieved.

18. L395- was the physical activity data collected using a stratified random sampling technique to ensure equal number of girls and boys? It does not appear so, but I think this would be important based on established sex-difference in this outcome.

As there were five students per class, it was not possible at the individual level within each class to achieve an equal number of boys and girls being assessed. However, through the use of a random number generator, we did a 2/3 split (either 2 girls and 3 boys or 2 boys and 3 girls). While we acknowledge that there is indeed an established sex-difference in these outcomes, examining such differences was not a key research question within the present manuscript. 

19. L491- Wouldn’t sex be a more appropriate term here?

Our preference is to keep the term gender. As you will see in supplementary materials, we asked students to self-identify their gender, not their sex. As such, for consistency, we will keep the same terminology used in the data collection.

20. L528- Please double-check for consistent language- fundamental movement skills, motor skills, motor competence, and motor skill outcomes are all used throughout this manuscript.

Thank you for bringing this to our attention. For consistency, we have used the term “fundamental movement skill(s)” throughout the manuscript. The only time “motor” is used is specifically when referencing the “Test of Gross Motor Development-3rd edition” which is how we assessed the process outcomes of two fundamental movement skills, kicking and overhand throwing. 

21. L535- if the interaction was non-significant, why is it re-included in the model and then interpreted (note: this appears to be further included as a finding in the discussion L629-631). 

The Reviewer brings up a very good point and as such we have rerun our analyses without the inclusion of this non-significant effect. Results in Table 5 have been updated and we have removed the Discussion points out of the manuscript.

22. Table 5- Throwing and kicking speed are well defined in the methods, but throw-catch combination score, throw components, and kick components are not clear. I am assuming that “components” is synonymous with TGMD or process. If that is the case, please revise the language in table 5 to align with the text.

Yes, the Reviewer is correct in their interpretation. We have now amended the language used in Table 5 to clearly articulate the process scores for throwing and kicking. Table 5 now lists “Throw Process Scores (as assessed by the Test of Gross Motor Development-3rd edition)” and “Kick Process Scores (as assessed by the Test of Gross Motor Development-3rd edition)”. 

23. L576- Please confirm that five-session is correct. L255 mentions 7.

We have amended the reference to the number of sessions in the Discussion to reflect that there were 7, not five sessions in total for this program. There were four lessons that focused on curriculum and three sessions that provided the peer-leaders (Grade 6/7 students) with practical opportunities to practice their teaching of the movement skill lessons. This change can now be seen on line 605.

Ultimately, we did not find evidence that the primary outcome, teacher rated transformational leadership of Grade 6/7 students could be changed by a seven-session leadership learning program and the subsequent implementation of a 10-week peer-led program.

24. It would be nice to have some greater depth in the discussion relating to other literature. For example, how do the null findings relate to other literature using similar train-the-trainer approaches (L592-596)? 

The author brings up a good point, as such we have added to our manuscript on lines 673 to 689. Previous work by our team and others has shown that previous peer-led interventions are capable of enacting meaningful physical activity behavior change. There is limited evidence in the peer-review literature that supports opportunities for meaningful change in motor skills, thus we require more studies before we can make definitive statements. When looking at other interventions, work by Lander et al. (2016) has shown that interventions where teachers are trained and delivered interventions can lead to meaningful improvements in both motor skills and physical activity. Overall, the null findings for most of our study outcomes could be seen as contradictory to previous literature. However, balanced against the novel design, more research will be needed to determine whether this study is unique in its null findings or whether scaled-up versions are capable of improving health and leadership behaviors. Relative to the pilot GLASS trial, there may have been program drift (i.e., the intervention was not delivered to the full extent or in as thorough a way). We now know the model of designing an intervention in this way is scalable, so moving forward we can tweak the program to try and ensure that sessions are more targeted on our outcomes of interest. With that said, as the reviewer rightly notes in their overall appraisal of the paper, we feel it is important to report such null findings, to ensure that researchers and practitioners have a fully informed understanding of the efficacy of various interventions in this setting. 

Citation:

Lander, N., Eather, N., Morgan, P. J., Salmon, J. & Barnett, L. M. Characteristics of Teacher Training in School-Based Physical Education Interventions to Improve Fundamental Movement Skills and/or Physical Activity: A Systematic Review. Sports Medicine 47, 135–161 (2016).

Existing evidence related to the use of a train-the-trainer model to deliver the intervention could be described as mixed. Thus, while the current study has null findings, this is not out of the ordinary. Indeed, a recent review of peer-led interventions found that while physical activity can be improved through such interventions. For the vast majority of behavioral, psychological, physiological, and leadership outcomes there were limited significant improvements (6). In contrast to these results, other intervention delivery modalities that use teachers or other trained facilitators appear to be more successful. Lander and colleagues (56) report that providing training to teachers, who subsequently lead and deliver school-based interventions, have shown improvements in physical activity behaviors and improved motor skills. Further still, programs such as ‘Supporting Children’s Outcomes using Rewards, Exercise and Skills' (SCORES) intervention (57) have shown that trained facilitators who receive ongoing support can lead to improvements in physical activity, movement skills, and other health-related outcomes. Yet, when scaled-up and less on-going support is provided, effect sizes were much smaller (58). Train-the-trainer strategies are important and necessary as they increase the feasibility of delivering programs. But who is trained and how much training they receive also plays a vital role for potential outcomes of the interventions.

25. Is it that the pilot study was flawed (L604) of that it was implemented at a more critical window of motor skill development and malleability? 

As we discuss on lines 612 to 647 we make the case for reasons we think the scale-up from the GLASS study to the current PLPL study may have been flawed. Prior research supports the common finding that there is decreased effectiveness of scaled-up version of trials compared to smaller, previous iterations. See Beets et al., 2020 for a full explanation.

Beets MW, Weaver RG, Ioannidis JPA, Geraci, M, Brazendale K, Decker L, et al. Identification and evaluation of risk of generalizability biases in pilot versus efficacy/effectiveness trials: a systematic review and meta-analysis. Int J Behav Nutr Phys Act. 2020; 17(1). 

We do not have enough evidence in this study to derive insight into when and where a ‘critical window’ of opportunity may exist for the improvement of movement skills. This would require a study design that lasts multiple years and/or examines a larger age cohort of younger students. However, it should be noted that multiple systematic reviews and meta-analyses related to motor skill development support the fact that motor skills can be improved at all ages throughout childhood. This would support that skill performance is malleable at any age and not specific to one point in developmental time. This would then suggest that a scaled-up version of our intervention should hypothetically work at any age. We have provided specific references below in support of this. 

Logan, S. W., Robinson, L. E., Wilson, A. E., & Lucas, W. A. (2012). Getting the fundamentals of movement: a meta‐analysis of the effectiveness of motor skill interventions in children. Child: care, health and development, 38(3), 305-315.

Van Capelle, A., Broderick, C. R., van Doorn, N., Ward, R. E., & Parmenter, B. J. (2017). Interventions to improve fundamental motor skills in pre-school aged children: A systematic review and meta-analysis. Journal of Science and Medicine in Sport, 20(7), 658-666.

Morgan, P. J., Barnett, L. M., Cliff, D. P., Okely, A. D., Scott, H. A., Cohen, K. E., & Lubans, D. R. (2013). Fundamental movement skill interventions in youth: A systematic review and meta-analysis. Pediatrics, 132(5), e1361-e1383.

Wick, K., Leeger-Aschmann, C. S., Monn, N. D., Radtke, T., Ott, L. V., Rebholz, C. E., ... & Kriemler, S. (2017). Interventions to promote fundamental movement skills in childcare and kindergarten: a systematic review and meta-analysis. Sports Medicine, 47(10), 2045-2068.

As stated on lines 657 to 672, the null motor skill findings could be the result of assessing a limited number of skills while the programming itself focused on a wider array of skills. It could also be that since there was a change in the implementation mechanism in this randomized controlled trial where the research team is one step removed compared to the pilot study (ultimately to support potential scale-up), these null findings could be expected. 

26. What longitudinal data is available on motor skills and psychological constructs measured (L649)?

Lastly, we mention that longitudinal studies would be needed to try and support a possible “critical window” hypothesis. Given that this study was an experimental design, we do not find it appropriate to cover information related to longitudinal measurement of motor skills and psychological constructs in the Discussion. We would refer the Reviewer to more appropriate (and recent) publications on this topic, such as the systematic review conducted by Barnett and colleagues published in Sports Medicine. This reference is provided below. 

Barnett, L. M. et al. Through the Looking Glass: A Systematic Review of Longitudinal Evidence, Providing New Insight for Motor Competence and Health. Sports Med 52, 875–920 (2022).

25. L613- What is FMS?

FMS refers to “fundamental movement skills”. This acronym has been specified in the third paragraph of our Background/Introduction and can be seen on line 101. 

Editorial Comments

Thank you for bringing to this our attention. We have carefully reviewed the provided style requirements and made the subsequent changes within the manuscript. 

2.Thank you for stating the following in the Funding Section of your manuscript:

“Funding for this project is provided by the Social Sciences and Humanities Research Council of Canada (SSHRC) [Insights Grant #435-2017-0268]. The funding body (SSHRC) had no role in the design, data collection, analysis, interpretation of data, or the writing of the manuscript. DRL is supported by a National Health and Medical Research Council Senior Research Fellowship (APP1154507).”

“Author M.B. received funding to conduct this project through the Social Sciences and Humanities Research Council of Canada (SSHRC; https://www.sshrc-crsh.gc.ca/home-accueil-eng.aspx) Insights Grant #435-2017-0268. Author DRL is supported by a National Health and Medical Research Council Senior Research Fellowship (APP1154507; https://www.nhmrc.gov.au/). The funding bodies, in both instances, had no role in the design, data collection, analysis, interpretation of data, or the writing of the manuscript.”

As requested, we have removed all funding information from the manuscript. As such, all funding related to this project is currently updated in the “Funding Information” section of the online submission portal. This includes as award from the Social Sciences and Humanities Research Council, as well as an investigator award from the National Health and Medical Research Council. 

3.We note that you have indicated that data from this study are available upon request. PLOS only allows data to be available upon request if there are legal or ethical restrictions on sharing data publicly. For more information on unacceptable data access restrictions, please see http://journals.plos.org/plosone/s/data-availability#loc-unacceptable-data-access-restrictions.

We have now included the deidentified data as part of our resubmission. As such, there is no need for addressing the above comment, as data will be publicly available for individuals. 

We have provided two files, one for peer leaders and one for younger Grade 3/4 students that can be used to replicate all findings. We have provided each of these files as a supplementary file. 

4.We note that you have stated that you will provide repository information for your data at acceptance. Should your manuscript be accepted for publication, we will hold it until you provide the relevant accession numbers or DOIs necessary to access your data. If you wish to make changes to your Data Availability statement, please describe these changes in your cover letter and we will update your Data Availability statement to reflect the information you provide.

Thank you. Again, as we are providing the data as supplementary files there is no longer a need for a Data Availability statement, as everyone will have access to the deidentified data set once accepted for publication.

---

## [Decision Letter · Decision Letter 1]

26 Dec 2022

Evaluation of the Peer Leadership for Physical Literacy Intervention: A Cluster Randomized Controlled Trial

PONE-D-22-11797R1

Dear Dr. Hulteen,

We’re pleased to inform you that your manuscript has been judged scientifically suitable for publication and will be formally accepted for publication once it meets all outstanding technical requirements.

Kind regards,

Venkat Rao Vishnumolakala

Academic Editor

PLOS ONE

Additional Editor Comments (optional):

Season's Greetings!

I sincerely thank you for considering PLOS One for your manuscript and patiently waiting for the outcome of the peer-review process. Please address the minor concerns raised by the reviewer.

Best Wishes,

Ven

Reviewers' comments:

Reviewer's Responses to Questions

**Comments to the Author**

1. If the authors have adequately addressed your comments raised in a previous round of review and you feel that this manuscript is now acceptable for publication, you may indicate that here to bypass the “Comments to the Author” section, enter your conflict of interest statement in the “Confidential to Editor” section, and submit your "Accept" recommendation.

Reviewer #1: All comments have been addressed

2. Is the manuscript technically sound, and do the data support the conclusions?

Reviewer #1: Yes

3. Has the statistical analysis been performed appropriately and rigorously? 

Reviewer #1: Yes

4. Have the authors made all data underlying the findings in their manuscript fully available?

Reviewer #1: Yes

5. Is the manuscript presented in an intelligible fashion and written in standard English?

Reviewer #1: Yes

6. Review Comments to the Author

Reviewer #1: Thank you again for your patience in this review. I know the authors and editors were naturally (and rightfully) anxious to move forward with this manuscript; however, the end of this term has been particularly challenging with personal and professional happenings. I sincerely apologize for my delay.

Thank you to the authors for your revisions. I believe the manuscript is stronger in its present form. I have no more major concerns with the document, but I do still have a few minor comments that should be addressed.

• L56, “student rated transformational leadership”. Is this a primary or secondary outcome? It would be helpful to have this listed in the descriptions of outcomes in L41-45.

• L60- Why is teacher adherence listed as the first point in the discussion when this was not a listed outcome (L41-45)?

• L125- Shouldn’t teams be possessive?

• Thank you for the increased description of PL. I think this helps situate the research.

• L250- Peer Leadership Intervention- Were FMS ever taught to the teachers to teach to the peer leaders in the train-the-trainer model?

• L404- Interesting that TGMD testing protocol was not followed but children were scored on this assessment. Perhaps this should be mentioned in the discussion

• L411- Include a statement that both RAs coded a unique set of children, and there was no overlap between coders.

• L425- Typo. Change to “much”.

• L429- What constitutes a catch? Could a trap motion be a catch?

• Table 2- Why is MVPA nested under Movement Skill Competence?

• L571- Please remove “also”.

• L572- Missing parenthesis

• For clarity, I suggest making separate tables for Grade 3/4 psychological outcomes, FMS, and PA. Right now, it is odd to have the FMS reported with psychological outcomes in the text but with PA in the table.

7. PLOS authors have the option to publish the peer review history of their article (what does this mean?). If published, this will include your full peer review and any attached files.

Reviewer #1: No

---

## [Editor Report · Acceptance letter]

7 Feb 2023

PONE-D-22-11797R1 

Evaluation of the Peer Leadership for Physical Literacy Intervention: A Cluster Randomized Controlled Trial 

Dear Dr. Hulteen:

I'm pleased to inform you that your manuscript has been deemed suitable for publication in PLOS ONE. Congratulations! Your manuscript is now with our production department. 

Kind regards, 

on behalf of

Dr. Venkat Rao Vishnumolakala 

Academic Editor

PLOS ONE